# MOAI: Module-Optimizing Architecture for Non-Interactive Secure Transformer Inference

**Linru Zhang**[1], **Xiangning Wang**[1], **Jun Jie Sim**[2], **Zhicong Huang**[3], **Jiahao Zhong**[1],
**Huaxiong Wang**[1], **Pu Duan**[2], **Kwok-Yan Lam**[1]
[1]Nanyang Technological University, [2]Ant International, [3]Ant Group

## Abstract

Privacy concerns have been raised in Large Language Models (LLM) inference when models are deployed in Cloud Service Providers (CSP). Homomorphic encryption (HE) offers a promising solution by enabling secure inference directly over encrypted inputs. However, the high computational overhead of HE remains a major bottleneck. To address this challenge, we propose MOAI, an efficient HE-based, non-interactive framework for secure transformer inference. MOAI gains significant efficiency improvement from: (1) a novel evaluation flow that combines column and diagonal packing with consistent strategies across all layers, eliminating expensive format conversions. (2) rotation-free algorithms for Softmax and LayerNorm that significantly reduce the number of costly HE rotations, removing 2448 HE rotations in BERT-base inference. (3) Column packing removes rotations in plaintext–ciphertext matrix multiplications and interleaved batching further reduces the rotations in ciphertext–ciphertext matrix multiplications. MOAI uses at least 1.7x fewer HE rotations compared to the state-of-the-art works across all matrix multiplications of BERT-base. As a result, We achieve a 52.8% reduction in evaluation time compared to the state-of-the-art HE-based non-interactive secure transformer inference, THOR (Moon et al., CCS'25). We then apply MOAI on the Powerformer's framework and achieve a 55.7% reduction in evaluation time compared to Powerformer (Park et al., ACL'25), which approximates Softmax and LayerNorm with simpler functions in transformer and proposes HE-based non-interactive transformer inference. We report an amortized time of 2.36 minutes per input on a single GPU environment. We show the extendibility by applying MOAI in LLaMA-3-8B. Our implementation is publicly available as open source.

## 1 Introduction

The Transformer architecture Vaswani et al. (2017) powers many NLP tasks but requires heavy computation and memory, motivating deployment on cloud service providers (CSP). There are growing concerns about potential privacy leakage when clients upload sensitive information to CSP.

Fully homomorphic encryption (FHE) Gentry (2009) enables direct, non-interactive computation over encrypted data, making it a promising primitive for protecting clients' data privacy. In FHE-based AI-as-a-Service works Bourse et al. (2018); Gilad-Bachrach et al. (2016); Mishra et al. (2020); Lam et al. (2024); Zhang et al. (2024b), a client encrypts its input data and uploads the ciphertexts to a cloud service provider (CSP), which performs model evaluation without any decryptions. The CSP then returns the encrypted inference results to the client, ensuring that it learns neither the client's inputs nor the inference outputs. However, the high computational overhead of homomorphic evaluation significantly limits the practicality of FHE-based inference, especially in scenarios that a large number of inputs has to be processed such as document classification, a representative application of encoder-only models like BERT Devlin et al. (2019).

**Related work.** Some works Chen et al. (2022); Pang et al. (2024); Lu et al. (2023); Dong et al. (2023); Zeng et al. (2025) targeting interactive privacy-preserving transformer inference, usually on

BERT, relied on a combination of FHE and MPC that splits the user's data into shares and distributes a share to the server.

The first non-interactive solution NEXUS, based purely on FHE, was proposed by Zhang et al. (2024a) (NDSS '25). They proposed several FHE-friendly matrix multiplications and approximations for activation functions. However, NEXUS does not provide an end-to-end inference solution. NEXUS only reports performance as an aggregation of microbenchmarks. In addition, due to inconsistent packing formats, its algorithms cannot be directly composed. As also observed by Lim et al. (2025); Moon et al. (2024), the conversions between these formats during secure inference are not explicitly explained and the time cost of format conversions is not included in the reported performance. LEAF Zhang et al. (2025) leverages coefficient-encoding matrix multiplication and functional bootstrapping for inference, but focuses only on the feedforward block rather than providing an end-to-end solution. The state-of-the-art HE-based non-interactive solution THOR Moon et al. (2024) (CCS'25) achieved end-to-end secure transformer inference. THOR provided inference on 128 tokens with the BERT-base model and reported 10 minutes per input on a single GPU. Format conversions are required in THOR to complete ciphertext-ciphertext matrix multiplications. These solutions do not modify the transformer components such as Softmax and LayerNorm, and thus avoid the need for re-training or fine-tuning.

Another line of research focuses on adapting transformers to FHE-friendly functions. Zimerman et al. (2024b) replaced non-polynomial functions (e.g., Softmax) with polynomial alternatives. Power-Softmax Zimerman et al. (2024a) replaced the exponential function in Softmax with a power function and applied conventional fine-tuning. The state-of-the-art solution in this direction, Powerformer Park et al. (2024b) (ACL'25) replaced Softmax and LayerNorm in the BERT-base model with power and linear functions, achieving 5.74 minutes per input on a single GPU. Although this line of work offers improved efficiency, it requires re-training through knowledge distillation, which limits its generalizability to other transformer architectures.

**Our contributions.**    We focus on designing a plug-and-play FHE framework that can be seamlessly integrated into existing transformer architectures. Similar to NEXUS and THOR, we do not modify any transformer components to avoid the need for re-training through knowledge distillation.

In this paper, we present MOAI: **M**odule-**O**ptimizing **A**rchitecture for non-interactive secure transformer **I**nference. MOAI is a FHE-based. We apply MOAI on the BERT-base model and report an amortized time of 2.36 minutes per input on a single GPU environment. We also extend our methods to decoder-only transformers, such as LLaMA-3-8B. MOAI outperforms the state-of-the-art work THOR, achieving 52.8% reduction in evaluation time in the same environment. We also show the extendibility of MOAI by applying MOAI's evaluation flow to converted FHE-friendly transformer architectures. MOAI outperforms the state-of-the-art work Powerformer, achieving 55.7% reduction in evaluation time when applying MOAI to Powerforer's FHE-friendly modifications. Our techniques can be summarized as follows:

(1) **Consistent matrix packing strategies to avoid extra format conversion.**    The input ciphertexts and the evaluation of $Q, K, V$ are in column packing. Then the output of our FHE evaluation of $QK^T$ is in diagonal packing, which matches the input format of our FHE evaluation of Softmax. After evaluating Softmax in diagonal packing, a ciphertext - ciphertext matrix multiplication (for $\mathrm{softmax}(QK^\intercal/\sqrt{d'})V$) takes input of one matrix encrypted in diagonal packing $(\mathrm{Enc}(\mathrm{softmax}(QK^\intercal/\sqrt{d'})))$ and one matrix encrypted in column packing $(\mathrm{Enc}(V))$ and the output matrix is encrypted in column packing. Resting parts such as feedforward layers are all in column packing, and are ready to be passed to the next transformer layer without format conversion. The details of the evaluation flow is in Figure 2. Our evaluation flow ensures the consistent packing formats across all layers to avoid expensive format conversion between layers. We also remove the need of performing encrypted matrix transpose separately. In THOR, additional format conversions and transpose are necessary.

(2) **Rotation-free Softmax and LayerNorm evaluations.**  Evaluating Softmax and LayerNorm in NEXUS and THOR includes a set of FHE rotations to compute a summation of data packed in a ciphertext. HE rotation is known as a computation intensive operation which affects the efficiency (e.g., Halevi & Shoup (2018)). We propose rotation-free Softmax and LayerNorm evaluation algorithms which eliminate rotation by putting the inputs to the summation in the same slot position of different ciphertexts. These algorithms save 2448 HE rotations in BERT-base inference.

(3) **Minimizing HE rotations in matrix multiplication.** Column packing is adopted to remove HE rotations in plaintext-ciphertext matrix multiplication. Interleaved batching is applied to further reduces the rotations in ciphertext–ciphertext matrix multiplications. MOAI achieves at least 1.7x fewer HE rotations than the state-of-the-art works across all matrix multiplications of a transformer.

## 2 PRELIMINARIES

We use $[n]$ to represent the set $\{0, 1, ..., n-1\}$. We use bold lowercase letters for row vector $\mathbf{v} \in \mathbb{R}^n$ where $\mathbf{v} = [v_0, v_1, ..., v_{n-1}]$. We use $[v_i]_{i\in[n]}$ to represent the vector $\mathbf{v}$. Column vectors are represented by $\mathbf{v}^\mathsf{T}$. We define the scalar product for $c \in \mathbb{R}$ and $\mathbf{v} \in \mathbb{R}^n$, let $c\mathbf{v} = [cv_i]_{i\in[n]}$. We also define addition and Hadamard product (entry-wise multiplication) between two vectors with dimension $n$ as follows: $\mathbf{v} + \mathbf{u} := [v_i + u_i]_{i\in[n]}, \mathbf{v} \otimes \mathbf{u} := [v_i u_i]_{i\in[n]}$. We use $\text{Rot}_k(\mathbf{v})$, for $0 \le k < n$, to denote the vector $[v_k, ..., v_{n-1}, v_0, ...v_{k-1}]$, a *left rotation* of $\mathbf{v}$. For $-n < k \le 0$, $\text{Rot}_k(\mathbf{v})$ denotes a right rotation.

We use uppercase letters to denote a matrix $A \in \mathbb{R}^{m \times d}$ where $A = (a_{ij})_{i\in[m],j\in[d]}$. We use $\mathbf{a_j}^\mathsf{T}$ to represent the $j$-th column of $A$, and we can write $A = (\mathbf{a_0}^\mathsf{T} \parallel \mathbf{a_1}^\mathsf{T} \parallel \cdots \parallel \mathbf{a}_{d-1}^\mathsf{T})$. Further, if $A$ is a square matrix with size $m \times m$, define $\text{Diag}_i(A)$ as $A$'s $i$-th (upper) diagonal for $i \in [m]$: $\text{Diag}_i(A) := [a_{0,i}, a_{1,i+1}, ..., a_{m-i-1,m-1}, a_{m-i,0}, ..., a_{m-1,i-1}]$.

### 2.1 FULLY HOMOMORPHIC ENCRYPTION - CKKS SCHEME

Fully Homomorphic Encryption (FHE) enables computation on encrypted data. The CKKS scheme Cheon et al. (2017) supports approximate arithmetic over complex vectors. We briefly describe CKKS and refer to Appendix I and Cheon et al. (2017) for details.

Let $N$ be the degree of a polynomial ring, whose value is a power of $2$. $Q$ be a product of distinct primes: $Q = \prod_0^L q_i$, where $L$ is called the *level* of the ciphertext. A fresh ciphertext starts from level $L$ and drops to $L-1$ after a homomorphic multiplication. When remaining level is 0, decryption fails. The bootstrapping process Cheon et al. (2018) can refresh the ciphertext and restore the remaining level, enabling continued evaluations. The CKKS scheme supports SIMD-style computation by encoding $N/2$ messages into a single ciphertext, enabling parallel homomorphic operations across all slots. Let $\mathbf{v}, \mathbf{v}'$ be two vectors in $\mathbb{C}^{N/2}$ and $\text{ct} = \text{Enc}(\mathbf{v})$ and $\text{ct}' = \text{Enc}(\mathbf{v}')$. The operations supported by the CKKS scheme are as follows:

$\text{Add}(\text{ct}, \text{ct}')$. Return a ciphertext that encrypts the vector sum $\mathbf{v} + \mathbf{v}'$.
$\text{Mult}(\text{ct}, \text{ct}')$. Return a ciphertext that encrypts the Hadamard product $\mathbf{v} \otimes \mathbf{v}'$.
$\text{PlainMult}(\text{ct}, \mathbf{v}')$. Return a ciphertext that encrypts the Hadamard product $\mathbf{v} \otimes \mathbf{v}'$.
$\text{Rot}_k^{\text{HE}}(\text{ct})$. Return a ciphertext that encrypts $\text{Rot}_k(\mathbf{v}) = [v_k, ..., v_{N/2-1}, v_0, ...v_{k-1}]$.

Many previous works state that HE rotation is one of the slowest operations, and they optimize schemes by reducing the number of HE rotation (e.g., Halevi & Shoup (2018); Lu et al. (2021)). Experimental results in (Yang et al. (2024); Agulló-Domingo et al. (2025)) also indicate that HE rotation performs considerably slower, especially as the ciphertext modulus increases.

### 2.2 TRANSFORMERS AND FHE-BASED SECURE TRANSFORMER INFERENCE

The transformer architecture Vaswani et al. (2017) follows an encoder–decoder design. We focus on the encoder (e.g., BERT), commonly used as a benchmark. Each encoder has $L$ layers with multi-head self-attention, normalization, and feed-forward blocks. Core computations include matrix multiplication, Softmax, LayerNorm, and GELU. Further details are in Appendix B.

In FHE-based secure transformer inference, the model will take encrypted tokens as input and output encrypted results. The model is evaluated over encrypted data homomorphically, where the FHE evaluations include plaintext-ciphertext matrix multiplication (computing $Q, K, V$ and feedforward layer), ciphertext-ciphertext matrix multiplication (computing $QK^\mathsf{T}$ and $\text{softmax}(\cdot)V$, and non-linear evaluations such as Softmax, LayerNorm and GELU.

## 3 MATRIX PACKING, MULTIPLICATIONS AND INTERLEAVED BATCHING

### 3.1 MATRIX PACKING AND MULTIPLICATIONS

The SIMD capability of CKKS scheme allows multiple data to be packed into a single ciphertext, which offers an array-like paradigm to encode matrices for encryption. There are mainly three ways to encode an encrypted matrix in Halevi & Shoup (2014), namely row, column, diagonal packing. The main difficulty in designing an encrypted matrix multiplication pipeline is that the encrypted matrix must be able to fit in and switch between various forms (e.g., the transpose $A^\mathsf{T}$). To this end, we propose to use both the column packing and diagonal packing to realize seamless transition for encrypted matrix multiplication.

**Definition 3.1** (Column Packing). *Given a matrix $X \in \mathbb{R}^{m \times d}$, the column packing of $X$ is a set of $d$ CKKS ciphertexts, denoted as $\mathrm{Enc}_{col}(X) := \left\{ \mathrm{Enc}(\mathbf{x}_0^\mathsf{T}), \mathrm{Enc}(\mathbf{x}_1^\mathsf{T}), ..., \mathrm{Enc}(\mathbf{x}_{d-1}^\mathsf{T}) \right\}$, where $\mathbf{x_j}^\mathsf{T}$ is the $j$-th column of $X$. We say that the ciphertext of $X$ is in **column packing**.*

**Definition 3.2** (Diagonal Packing). *Given a **square** matrix $X \in \mathbb{R}^{m \times m}$, the (upper) diagonal packing of $X$ are $m$ CKKS ciphertexts, denoted as $\mathrm{Enc}_{diag}(X) := \left\{ \mathrm{Enc}(\mathrm{Diag}_0(X)), \mathrm{Enc}(\mathrm{Diag}_1(X)), ..., \mathrm{Enc}(\mathrm{Diag}_{m-1}(X)) \right\}$, where $\mathrm{Diag}_j(X)$ is the $j$-th (upper) diagonal of $X$. We use $\mathrm{Diag}_j(X)$ as a **row** vector and say ciphertext of $X$ is in **diagonal packing**.*

Encrypted matrix-vector multiplication for both packing methods were proposed in Halevi & Shoup (2014). We give a brief description of the algorithms for encrypted matrix-vector multiplication for both packing methods. Let $C$ be a square matrix of dimensions $m \times m$ and $v$ be a vector of length $m$.

For column packing, denote $C$ with its $m$ columns as $\{\mathrm{Col}_0(C), \mathrm{Col}_1(C), ..., \mathrm{Col}_{m-1}(C)\}$. Halevi & Shoup (2014) first proposed an auxiliary algorithm `Replicate` that takes the vector $\mathbf{v}$ and return $m$-many vectors $\mathbf{v_i}$, each encrypting $v_i$, that is $\{\mathbf{v_i} = [v_i, v_i, \ldots, v_i]\}_{i \in [m]}$. The matrix-vector product is computed as $Cv = \sum \mathrm{Col}_i(C) \odot \mathbf{v_i}$. We can extend the matrix-vector product to matrix-matrix product by iterating over all columns of the right hand side matrix.

For diagonal packing, denote $C$ with its $m$ diagonals as $\{\mathrm{Diag}_0(C), \mathrm{Diag}_1(C), ..., \mathrm{Diag}_{m-1}(C)\}$. The matrix-vector product is computed as $Cv = \sum \mathrm{Diag}_i(C) \odot \mathrm{Rot}(\mathbf{v}, i)$. A subsequent improvement in Halevi & Shoup (2021) reduced the number of rotations needed to $O(\sqrt{m})$ by using the baby-step giant-step method (see Appendix C.1 for details).

We describe our HE matrix multiplications in both column and diagonal packing:

**Case I:** Calculating $\mathrm{Enc}(Q), \mathrm{Enc}(K), \mathrm{Enc}(V) \in \mathbb{R}^{m \times d'}$ with $\mathrm{Enc}(X)W_i + \mathbf{1}^\mathsf{T}\mathbf{b}_i$ for $i \in \{Q, K, V\}$. We encrypt $X$ in column packing while the weights and biases are not encrypted. We note that since the weights matrices are not encrypted, the matrix product can be interpreted as concatenations of linear combinations of $X$. $XW = [\sum_{i \in [d']} w_{i,0}\mathbf{x}_0^\mathsf{T} \parallel \sum_{i \in [d']} w_{i,1}\mathbf{x}_1^\mathsf{T} \parallel \cdots \parallel \sum_{i \in [d']} w_{i,d'-1}\mathbf{x}_{d'-1}^\mathsf{T}]$. The detailed algorithm is given as Algorithm 2 in Appendix C.2.

**Case II:** Calculating $\mathrm{Enc}(QK^\mathsf{T})$. We remind the reader here that $Q$ and $K$ are encrypted in column packing as per the result of Case I. We then apply the following lemma that allows us to compute the matrix product $QK^\mathsf{T}$ where both $Q$ and $K$ are encrypted in column packing, without transposing $K$.

**Lemma 3.1.** *Suppose $Q, K \in \mathbb{R}^{m \times d'}$, so $QK^\mathsf{T}$ is square matrix with size $m \times m$. Then for $j \in [m]$: $\mathrm{Diag}_j(QK^\mathsf{T}) = \sum_{i=0}^{d'-1} \mathbf{q}_i \otimes Rot_j(\mathbf{k}_i)$, where $\mathbf{q}_i$ and $\mathbf{k}_i$ are the $i$-column of $Q$ and $K$ respectively.*

The result is a matrix encrypted in diagonal packing. The detailed algorithm is given as Algorithm 3 in Appendix C.2.

**Case III:** Calculating $\mathrm{Enc}(\mathrm{softmax}(QK^\mathsf{T}/\sqrt{d'})V)$. The encrypted matrix $V$ is encrypted in column packing, as per Case I. For now, we briefly mention that applying the Softmax function on a diagonally packed matrix will return a diagonally packed matrix. Thus, $\mathrm{softmax}(QK^\mathsf{T}/\sqrt{d'})$ is encrypted in diagonal packing. For this matrix product, we can apply the matrix-vector product for diagonally packed matrices described earlier Halevi & Shoup (2014; 2021) over all encrypted columns of $V$. The detailed algorithm is given as Algorithm 4 in Appendix C.2.

## 3.2 INTERLEAVED BATCHING

In column packing (Definition 3.1) and diagonal packing (Definition 3.2), each ciphertext encrypts a vector of dimension $m$, i.e., the sequence length of input. Meanwhile, the number of slots $N/2$ in the CKKS scheme is usually $2^{15}$. So batching multiple vectors into one ciphertext will reduce the amortized cost. Interleaved batching, previously discussed in Aharoni et al. (2023); Adir et al. (2024) for evaluating convolutions in neural networks, is adopted in MOAI to further reduce the number of HE rotations in matrix multiplications.

Assume $m|(N/2)$. Let $N/(2m)$ vectors $\{\mathbf{x}^{(r)} \in \mathbb{R}^m\}_{r \in [N/(2m)]}$ be:

$$\mathbf{x}^{(0)} = [x_0^{(0)}, x_1^{(0)}, ..., x_{m-1}^{(0)}], ..., \mathbf{x}^{(N/(2m)-1)} = [x_0^{(N/(2m)-1)}, x_1^{(N/(2m)-1)}, ..., x_{m-1}^{(N/(2m)-1)}],$$

*interleaved batching* is defined by packing the the first entry of each vector from $\{\mathbf{x}^{(r)} \in \mathbb{R}^m\}_{r \in [N/(2m)]}$, then the second entry, and at last the last entry. We use $\tilde{\mathbf{x}} \in \mathbb{R}^{N/2}$ to represent the interleaved batching of $N/(2m)$ vectors in $\mathbb{R}^m$.

$$\tilde{\mathbf{x}} := [x_0^{(0)}, x_0^{(1)}, x_0^{(2)}, ..., x_0^{(N/(2m)-1)}, ...x_{m-1}^{(0)}, x_{m-1}^{(1)}, x_{m-1}^{(2)}, ..., x_{m-1}^{(N/(2m)-1)}]. \tag{1}$$

We use Lemma 3.2 to address the its compatibility with rotation and include its proof in Appendix D.2. The naive batching method in existing works uses two HE rotations to do one "internal rotation", which only shifts slots within each vector rather than across all slots in a ciphertext. MOAI achieves the same functionality by only one HE rotation, thereby reducing half of HE rotations.

**Lemma 3.2.** *Let $\tilde{\mathbf{x}} \in \mathbb{R}^{N/2}$ be the interleaved batching vector of $\{\mathbf{x}^{(r)} \in \mathbb{R}^m\}_{r \in [N/(2m)]}$ given by Eq. (1). Then $Rot_{jN/(2m)}(\tilde{x})$ is the interleaved batching vector of $\{Rot_j(\mathbf{x}^{(r)})_{r \in [N/(2m)]}\}$.*

Finally, we define interleaved column/diagonal packing by merging interleaved batching with our matrix packing methods directly. Please refer to Appendix D.3.

## 3.3 COMPLEXITY COMPARISON

We provide a comparison of our matrix multiplications with matrix multiplications in prior FHE-based secure transformer inference works NEXUS Zhang et al. (2024a)(NDSS'25), THOR Moon et al. (2024)(CCS'25), and Powerformer Park et al. (2024b)(ACL'25). We focus on the total number of HE rotations, which is the most expensive underlying HE operations as discussed in Section 2.1.

In Table 1, we report the total number of HE rotations in all matrix multiplications of a transformer.[1] MOAI requires only 9648 HE rotations in all matrix multiplications in BERT model, achieving $22.9$x, $2.3$x, and $1.7$x fewer HE rotations than NEXUS, THOR, and Powerformer, respectively.

For consistency with prior work, we include a comparison if the total number of key switches in Appendix E. Key switch is the key component in the relinearization process of ciphertext multiplication and the HE rotation. MOAI requires only 834 key switches for all matrix multiplications in a single multi-head attention block, MOAI achieves the least number of key switches.

## 4 NON-LINEAR FUNCTION EVALUATION

In this section we propose optimized evaluations of non-linear functions. Our novelty is that we do not require any rotations in evaluations of Softmax and LayerNorm. This greatly improves the speed of our algorithms. Table 2 gives a comparison between ours and NEXUS Zhang et al. (2024a) (NDSS'25) and THOR Moon et al. (2024) (CCS'25). When substitute BERT's parameters into the equation, our rotation-free Softmax and LayerNorm save 2448 HE rotations in total.

**Softmax.** The Softmax function appears in $\mathrm{softmax}(QK^\intercal/\sqrt{d'})V$, but we eliminate the $\sqrt{d'}$ term by folding it into $W_K$, saving one CKKS level. The input of Softmax is in diagonal packing, and we

---

[1] 1)For NEXUS, results are taken from Zhang et al. (2024a) and Table 4 in Moon et al. (2024), and normalized by batch size. NEXUS does not specify how it evaluates the third line, so we exclude this result for NEXUS. 2)For THOR, results are taken from Table 4 in Moon et al. (2024), include counts only from HE rotations. 3) For Powerformer, the attention block's HE rotation count in our table is computed by Algorithm 14 in Park et al. (2024b).

Table 1: Comparison of number of HE rotations in matrix multiplications. For short, we use $\sigma$ to denote Softmax and omit $\sqrt{d'}$. $t$ is the intermediate size of the feed-forward layer. $H$ is the number of heads. $c = 16$ is from THOR. $k = 64$ is from Powerformer.

| Operation | Function | NEXUS #Rot$^{\text{HE}}$ | THOR #Rot$^{\text{HE}}$ | Powerformer #Rot$^{\text{HE}}$ | **Our MOAI #Rot$^{\text{HE}}$** |
|---|---|---|---|---|---|
| Attention block | $\text{Enc}(X)W_*^{(h)}, * = Q, K, V$ $X \in \mathbb{R}^{m \times d}, W_*^{(h)} \in \mathbb{R}^{d \times d'}, h \in [H]$ | $\frac{2md}{N/(2m)}$ | $\frac{m}{2} + 3(H-2)\frac{d'}{c} + \frac{3d'}{c}$ | $\sqrt{\frac{5}{9}} \cdot 2\sqrt{\frac{md(3d)}{N/2}}$ | 0 |
| | $\text{Enc}(Q^{(h)})\text{Enc}((K^{(h)})^{\intercal})$ $Q^{(h)}, K^{(h)} \in \mathbb{R}^{m \times d'}$ | $\frac{md'H}{N/(4m)}$ | $\frac{m}{2c}(\frac{m}{2}+5) + \frac{m}{2}$ $+ \frac{m}{8}(\log c + 3)$ | $(16 + 14\sqrt{2})\sqrt{k} + 3k$ | $\frac{md'}{N/(2m)} \cdot H$ |
| | $\text{Enc}(\sigma(Q^{(h)}(K^{(h)})^{\intercal}) \text{Enc}(V^{(h)})$ $\sigma(Q^{(h)}(K^{(h)})^{\intercal}) \in \mathbb{R}^{m \times m}, V^{(h)} \in \mathbb{R}^{m \times d'}$ | N/A | $m(\log c + 3)/4 + m\frac{m+2}{4c}$ | $(18 + 10\sqrt{2})\sqrt{k} + 3k$ | $\frac{d'(\sqrt{m}+m)}{(N/(2m))} \cdot H$ |
| | $\text{Enc}((X^{(0)} \parallel ... \parallel X^{(H-1)}))W_{fc0}$ $X^{(h)} \in \mathbb{R}^{m \times d}, W_{fc0} \in \mathbb{R}^{d \times d}$ | $\frac{2d^2}{N/(4m)}$ | $d'/2 + 2(H/2-1)m/c$ | $\sqrt{\frac{2}{3}} \cdot 2\sqrt{\frac{md^2}{N/2}}$ | 0 |
| FC1 | $\text{Enc}(X)W_{fc1} \ X \in \mathbb{R}^{m \times d}, W_{fc1} \in \mathbb{R}^{d \times t}$ | $\frac{2md}{N/(4m)}$ | $\frac{m}{2} + \frac{2t}{c}$ | $\sqrt{\frac{1}{2}} \cdot 2\sqrt{\frac{mdt}{N/2}}$ | 0 |
| FC2 | $\text{Enc}(X)W_{fc2} \ X \in \mathbb{R}^{m \times t}, W_{fc2} \in \mathbb{R}^{t \times d}$ | $\frac{2mt}{N/(4m)}$ | $\frac{m}{2} + \frac{2d}{c}$ | $\sqrt{\frac{1}{2}} \cdot 2\sqrt{\frac{mtd}{N/2}}$ | 0 |
| Total (Substitute BERT's parameters into the equation: $N = 2^{16}, m = 128, d = 768, d' = 64, H = 12$, 12 layers) | | $> 221184$ | 22224 | 16740 | 9648 |

Table 2: Comparison with NEXUS Zhang et al. (2024a), THOR Moon et al. (2024) and MOAI Softmax and LayerNorm.

| Function | NEXUS #Rot$^{\text{HE}}$ | THOR #Rot$^{\text{HE}}$ | **MOAI (ours) #Rot$^{\text{HE}}$** |
|---|---|---|---|
| softmax$(X \in \mathbb{R}^{m \times m})$ | $\lceil \log_2(m) \rceil 4m \cdot 2^{\lceil \log_2(m) \rceil}/N$ | $\lceil \log_2(m) \rceil 4m \cdot \cdot 2^{\lceil \log_2(m) \rceil}/N$ | **0** |
| LayerNorm$(X \in \mathbb{R}^{m \times d})$ | $\lceil \log_2(d) \rceil 4d \cdot 2^{\lceil \log_2(d) \rceil}/N$ | $\lceil \log_2(d) \rceil 4d \cdot 2^{\lceil \log_2(d) \rceil}/N$ | **0** |
| Total (Substitute BERT's parameters into the equation: $N = 2^{16}, m = 128, d = 768, H = 12$, 12 layers) | 2448 | 2448 | **0** |

want the output of the Softmax function to be in diagonal packing, as the input to the next module. We introduce a novel FHE Softmax algorithm (Algorithm 1, Figure 1) that preserves this packing format and eliminates the need for rotations, based on the key insight in Lemma 4.

**Lemma 4.1.** *Let $C \in \mathbb{R}^{m \times m}$ be a square matrix. The summation of columns equals to the summation of diagonals: $\sum_i \mathbf{c}_i^{\intercal} = \sum_i \text{Diag}_i(C)$.*

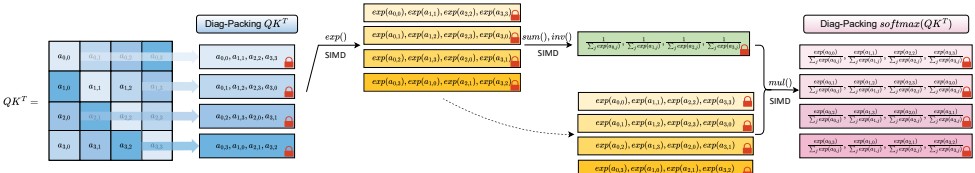

Figure 1: Our rotation-free Softmax evaluation whose input and output are both in diagonal packing.

**LayerNorm.** Note that the input of LayerNorm is in column packing. We first give the SIMD style mean and variance algorithm, given input as $\text{Enc}_{col}(X)$ where $X \in \mathbb{R}^{m \times d}$. For $\text{Enc}(mean(X))$, compute the summation of all encrypted columns of $X$ and then multiply $1/d$: $\text{sMult}(1/d, \sum_i \text{Enc}(\mathbf{x}_i^{\intercal}))$. For $\text{Enc}(var(X))$. First compute $\text{ct}_{x^2} := \text{Mult}(\text{Enc}(\mathbf{x}_i) - \text{Enc}(mean(X)), \text{Enc}(\mathbf{x}_i) - \text{Enc}(mean(X)))$, $i \in [d]$. Then compute $\text{sMult}(1/d, \sum_{i \in [d]} \text{ct}_{x^2})$. We can merge two scalar multiplication of $1/d$ to save one FHE level. Goldschmidt division algorithm Goldschmidt (1964); Qu & Xu is applied to obtain the inverse square root $(1/\sqrt{x})$ of variance. Please refer to Algorithm 8 for the detailed explanation in Appendix F.

**GELU.** For the encrypted GELU algorithm using Equation 2, it suffices to approximate $\tanh(\cdot)$ function by polynomial. The main challenge is the balance between the size of effective input interval and the polynomial degree. By Cheon et al. (2022), a degree-$\Omega(R)$ polynomial is required when using minimax approximation with fixed maximum error on the domain interval $[-R, R]$. In our work, we approximate $\tanh(\cdot)$ in Equation 2 using a degree-23 polynomial $P_{23}(x)$ on interval $[-20, 10]$ which is enough in our experiment. If the model needs larger input interval, we apply the *domain extension polynomial* $B_r(x) := \frac{x}{L^r} - \frac{4}{27B^2 L^{3r}} x^3$ from Cheon et al. (2022). Let $P(x)$ be the minimax polynomial of $\tanh(x)$ on $[-B, B]$. Then $P \circ B_0(x)$ can take input from a larger interval $[-LB, LB]$ for proper $L$. (Theorem 4 in Cheon et al. (2022)). This can be further extended to input interval

---

**Algorithm 1** HE Softmax evaluation: $\text{Enc}_{diag}(QK^\intercal) \rightarrow \text{Enc}_{diag}(\sigma(QK^\intercal))$

---

**Input:** $\text{Enc}_{diag}(QK^\intercal)$: Diagonal packing of encrypt matrix $QK^\intercal \in \mathbb{R}^{m \times m}$.
**Output:** $\text{Enc}_{diag}(\text{softmax}(QK^\intercal))$: Diagonal packing of encrypt matrix $\sigma(QK^\intercal) \in \mathbb{R}^{m \times m}$.
 1: Write $\text{Enc}_{diag}(QK^\intercal)$ as $\{\text{Enc}(\text{Diag}_0(QK^\intercal)), \text{Enc}(\text{Diag}_1(QK^\intercal)), ..., \text{Enc}(\text{Diag}_{m-1}(QK^\intercal))\}$.
 2: Evaluate $\text{Enc}(\exp(\text{Diag}_i(QK^\intercal)))$ by SIMD polynomial evaluation $(1 + x/2^r)^{2^r}$.
 3: Add ciphertexts to obtain $\text{ct}_{sum} = \text{Enc}(\sum_i \exp(\text{Diag}_i(QK^\intercal)))$.
 4: Evaluate FHE inverse algorithm on $\text{ct}_{sum}$ using the Goldschmidt division algorithm Goldschmidt (1964).
    Let the result be $\text{ct}_{sum^{-1}} := \text{Enc}(1/\sum_i \exp(\text{Diag}_i(QK^\intercal)))$.
 5: For $i \in [m]$, return $\text{Mult}(\text{ct}_{sum^{-1}}, \text{Enc}(\exp(\text{Diag}_i(QK^\intercal))))$ as the $i$-th component.

---

$[-L^s B, L^s B]$. We also point out that using piecewise polynomial in NEXUS Zhang et al. (2024a) to approximate GELU has precision problem. Please refer to Appendix F.2 for a detailed analysis.

## 5  MOAI: FULL FLOW

The name MOAI also comes from the monolithic human figures on Easter Island. They are huge and stable. They are similar to our scheme: large and stable end-to-end inference, and with very few "rotations". We show our full diagram in Figure 2 of the BERT-base-uncased model, which is implemented and test in Section 6.1. The proposed packing flow, evaluation algorithms and interleaved batching techniques are applicable to other transformer models and even decoder-only models such as Llama3-8B.

We illustrate the dimensions and the packing formats of encrypted matrices in each module. The packing formats include column packing (Col-packing, Definition 3.1) and diagonal packing (Diag-packing, Definition 3.2). CPMM stands for ciphertext-plaintext matrix multiplication, while CCMM stands for ciphertext-ciphertext matrix multiplications. As shown in Figure 2, MOAI does not need conversion between the two packing methods.

Regarding to the number of HE rotations, Table 1 and Table 8 show that MOAI requires only 9,648 HE rotations in the full flow, yielding $23.2\times$, $2.56\times$, and $1.46\times$ fewer HE rotations than NEXUS, THOR, and Powerformer, respectively. Unlike Powerformer, which bypasses HE rotations by replacing Softmax and LayerNorm, MOAI preserves both and eliminates all HE rotations with rotation-free evaluation algorithms.

Figure 2: The full diagram of MOAI. The parameters are from the BERT-base-uncased model, which has 12 transformer layers and 12 heads in each layer. $c$ is the number of classes.

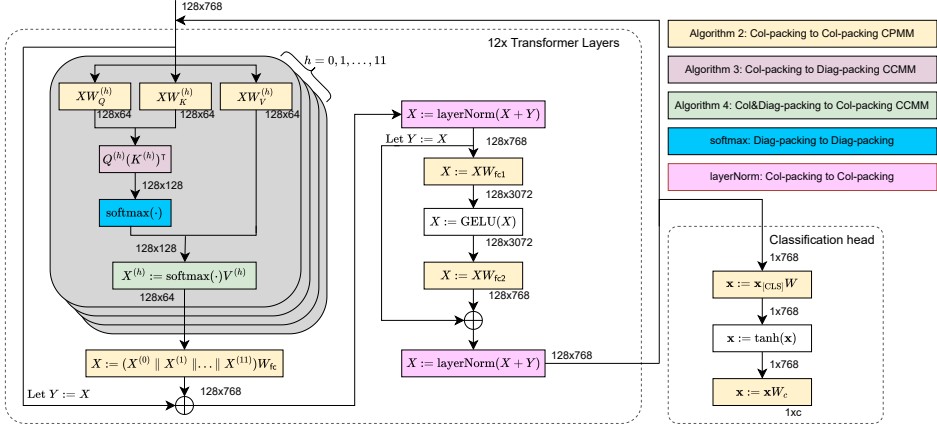

**About bootstrapping.**    Bootstrapping is an expensive operation that resets ciphertext noise, making its placement critical for performance. We make the following two observations: (1) A sufficiently accurate Softmax requires at least 10 iterations of the Goldschmidt division (line 4 in Algorithm 1, based on experimental results in Figure 5), which together with exponentiation and final multiplication

amounts to at least 20 layers (8 for $e^x$, 11 for division, and 1 for multiplication). (2) The runtime of matrix multiplication increases with noise level, as higher-level ciphertexts have larger moduli.

To address this, we perform bootstrapping on $\text{ct}_{sum}$ after the third line of Algorithm 1, refreshing it before Goldschmidt division. This reduces the Softmax depth from 20 to 10 layers, allowing the two preceding matrix multiplications to be executed at a lower level. As a result, the attention block starts from lower layers, and a single bootstrapping operation on one ciphertext substantially decreases both multiplication and overall runtime. The optimized Softmax algorithm is given in Algorithm 9.

## 6 EXPERIMENTAL RESULTS

**Implementation details.** We set the degree of the polynomial ring as $N = 2^{16}$, yielding $2^{15}$ slots in a ciphertext. We chose a 1743-bit ciphertext modulus of in accordance to the Homomorphic Encryption Standard Albrecht et al. (2021) to achieve 128-bit security. We apply the scale propagation technique Bossuat et al. (2021) to better manage the noise propagation and set the scale to $2^{46}$ in our implementation. Our codes are available here.[2]

*CPU implementation.* We implement MOAI in C++, utilizing the SEAL library SEAL for CKKS homomorphic encryption, FHE-MP-CNN[3] for CKKS bootstrapping, and OpenMP Dagum & Menon (1998) for multi-thread programming. All CPU results are evaluated on a Intel(R) Xeon(R) Platinum 8480+ CPU at 2.0 GHz with 56 cores.
*GPU implementation.* MOAI is implemented using the Phantom library[4] for CKKS encryption and bootstrapping. GPU results are evaluated on a single NVIDIA H200 GPU (end-to-end evaluation in Table 3) or a single NVIDIA A100 GPU (comparison with SOTAs in Table 4 and Table 5).

**Model and Dataset.** Same as SOTA works THOR Moon et al. (2024) and Powerformer Park et al. (2024b), we test MOAI's end-to-end performance on BERT-base model. It has 12 attention heads and 12 layers with an embedding dimension of 768. The input token length is 128. In evaluations, we use the pre-trained BERT-base-uncased model fine-tuned on SST-2, QNLI, and RTE tasks.[5]

**Microbenchmark.** Our optimized algorithms achieve speedups of up to $22\times$ for Softmax and $151\times$ for LayerNorm compared to prior works. We report the microbenchmark results in Appendix H.1.

### 6.1 END-TO-END INFERENCE EVALUATION

Table 3 presents the end-to-end performance (amortized for 256 inputs). The results are obtained from the execution of the BERT model with 12 transformer layers. For evaluating BERT model with input contains up to 128 tokens, we report an amortized time of 9.6 minutes per input on CPU and 2.36 minutes per input on GPU. Beyond encoder-only models, we extend our methods to decoder-only transformers, demonstrating them on LLaMA-3-8B inference and reporting results in Appendix H.2.

### 6.2 COMPARISON WITH SOTAS.

We compare MOAI with state-of-the-art works on a single NVIDIA A100 GPU, consistent with THOR Moon et al. (2024) and Powerformer Park et al. (2024b). For fair comparison, we also incorporate Powerformer's pooler and classification evaluation algorithm into our packing method (column packing) to account for total evaluation time. [6]

Similar to THOR, MOAI is designed as a plug-and-play FHE framework that can be seamlessly integrated into existing transformer architectures. This line of work does not modify the transformer components such as Softmax and LayerNorm, and does not need re-training or fine-tuning. Table 4

---

[2]CPU implementation: `https://github.com/dtc2025ag/MOAI`. GPU implementation: `https://github.com/dtc2025ag/MOAI_GPU`.

[3]https://github.com/snu-ccl/FHE-MP-CNN

[4]`https://github.com/encryptorion-lab/phantom-fhe`

[5]`https://github.com/huggingface/transformers/tree/main/examples/pytorch/text-classification`

[6]Similar to THOR and Powerformer, we do not compare end-to-end time with NEXUS Zhang et al. (2024a), since it only provides microbenchmark results. Its inconsistent packing formats require costly conversions that are neither described in the paper nor implemented in its open-source code, making end-to-end reproduction infeasible (see Moon et al. (2024); Lim et al. (2025)).

Table 3: Breakdown of MOAI encrypted inference, amortized over 256 inputs

| Component | Operation in each layer | Depth | Packing | (Data dimensions) × (Number of operations) | CPU time(s) | GPU time(s) |
|---|---|---|---|---|---|---|
| Attention | Pt-ct MatrixMul | $15 \to 14$ | Col → Col | $(\mathbb{R}^{128 \times 768} \times \mathbb{R}^{768 \times 64}) \times 36$ | 37.4 | 10.2 |
| | Ct-ct MatrixMul | $14 \to 13$ | Col → Diag | $(\mathbb{R}^{128 \times 64} \times \mathbb{R}^{64 \times 128}) \times 12$ | 40.3 | 8.55 |
| | Softmax | $13 \to 3$ | Diag → Diag | $\mathbb{R}^{128 \times 128} \times 12$ | 53.3 | 0.74 |
| | Ct-ct MatrixMul | $3 \to 2$ | Diag → Col | $(\mathbb{R}^{128 \times 128} \times \mathbb{R}^{128 \times 64}) \times 12$ | 1.4 | 0.26 |
| Self Output | Pt-ct MatrixMul | $2 \to 1$ | Col → Col | $(\mathbb{R}^{128 \times 768} \times \mathbb{R}^{768 \times 768})$ | 1.7 | 0.41 |
| | Bootstrapping | $1 \to 21$ | Col → Col | $\mathbb{R}^{128 \times 768}$ | 95.4 | 26.1 |
| | LayerNorm | $21 \to 1$ | Col → Col | $\mathbb{R}^{128 \times 768}$ | 0.6 | 0.05 |
| | Bootstrapping | $1 \to 21 \to 10$ | Col → Col | $\mathbb{R}^{128 \times 768}$ | 95.8 | 26.2 |
| Intermediate | Pt-ct MatrixMul | $10 \to 9$ | Col → Col | $(\mathbb{R}^{128 \times 768} \times \mathbb{R}^{768 \times 3072})$ | 44.1 | 12.7 |
| | GELU | $9 \to 2$ | Col → Col | $\mathbb{R}^{128 \times 3072}$ | 3.3 | 1.38 |
| Final | Pt-ct MatrixMul | $2 \to 1$ | Col → Col | $(\mathbb{R}^{128 \times 3072} \times \mathbb{R}^{3072 \times 768})$ | 7.1 | 2.5 |
| | Bootstrapping | $1 \to 21$ | Col → Col | $\mathbb{R}^{128 \times 768}$ | 98.8 | 26.1 |
| | LayerNorm | $21 \to 1$ | Col → Col | $\mathbb{R}^{128 \times 768}$ | 0.6 | 0.05 |
| | Bootstrapping | $1 \to 21 \to 15$ | Col → Col | $\mathbb{R}^{128 \times 768}$ | 94.8 | 26.07 |
| Total | | | | | **574.6** | **141.3** |

reports the layer-wise performance comparison between MOAI and THOR in the same environment with same ring dimension ($N = 2^{16}$) and slot size ($n = 2^{15}$). By reducing the number of HE rotations and adopting consistent packing strategies, MOAI achieves a 52.8% reduction in total time.

Furthermore, we apply MOAI's packing methods and evaluation algorithms to Powerformer which replaces Softmax and LayerNorm with FHE-friendly functions (BPmax and Batch LN) and retrains the model accordingly. Table 5 reports the layer-wise performance comparison between MOAI and Powerformer in the same environment, following Powerformer's parameter setting, i.e., $2^{15}$ slots in a ciphertext and 11 multiplicative levels before bootstrapping. MOAI achieves a 55.7% reduction in total evaluation time, demonstrating that our packing methods and evaluation algorithms can significantly improve the efficiency of retrained, FHE-friendly transformer architectures.

| Operation | Ours | THOR | Diff |
|---|---|---|---|
| Attention layer | 16.54 | 49.77 | 33.23 |
| Attention score | 14.07 | 16.25 | 2.18 |
| Softmax | 2.19 | 15.53 | 13.34 |
| Attention heads | 0.29 | 13.08 | 12.79 |
| Multi-head attention | 0.48 | 27.43 | 26.95 |
| LayerNorm1 | 0.15 | 7.13 | 6.98 |
| FC1 | 16.18 | 49.8 | 33.62 |
| GELU | 3.3 | 29.42 | 26.94 |
| FC2 | 2.88 | 49.19 | 46.31 |
| LayerNorm2 | 0.15 | 4.1 | 3.95 |
| Pooler & Classification | 0.7 | 2.7 | 2 |
| Bootstrappings | 227.84 | 337.86 | 110.02 |
| Total | 283.95 | 602.26 | 318.31 |
| Total w/o Pooler & Class. | 283.25 | 599.56 | 316.31 |

Table 4: Breakdown of execution time on a single A100 GPU compared to THOR

| Operation | Ours | Powerformer | Diff |
|---|---|---|---|
| Attention layer | 14.29 | 57.65 | 43.36 |
| Attention score | 6.61 | 28.76 | 22.15 |
| BPmax | 0.06 | 0.75 | 0.69 |
| Attention heads | 0.39 | 18.95 | 18.56 |
| Multi-head attention | 4.05 | 22.54 | 18.49 |
| Batch LN1 | 0.004 | 0.37 | 0.366 |
| FC1 | 4.79 | 59.21 | 54.42 |
| GELU | 5.2 | 8.31 | 5.83 |
| FC2 | 13.79 | 43.77 | 29.98 |
| Batch LN2 | 0.004 | 0.3 | 0.296 |
| Pooler & Classification | 0.7 | 0.2 | -0.5 |
| Bootstrappings | 102.5 | 103.72 | 1.22 |
| Total | 152.4 | 344.52 | 192.12 |
| Total w/o Pooler & Class. | 151.7 | 344.32 | 192.62 |

Table 5: Breakdown of execution time on a single A100 GPU compared to Powerformer

## 6.3 ACCURACY.

We analyze error propagation during encrypted evaluation by recording layer outputs and comparing them with plaintext results (Figure 3). The error grows over the first 6–7 layers but then stabilizes, indicating that it remains bounded. This shows that FHE evaluation does not cause significant accuracy loss. Further details on error variation within a layer are provided in Appendix H.3. Table 6 reports GLUE benchmark scores under both plaintext and encrypted inputs. The results indicate that the additional error introduced by FHE has only a negligible effect on inference accuracy.

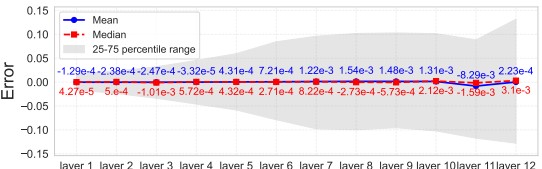

Figure 3: Error analysis of 12 layers' output

Table 6: Scores on the GLUE benchmarks

| Dataset | SST-2 | QNLI | RTE |
|---|---|---|---|
| Unencrypted input | 93.9 | 90.7 | 65.7 |
| Encrypted input | 93.8 | 90.5 | 65.5 |
| Match count | 1818 | 5437 | 2959 |
| Mismatch count | 4 | 27 | 42 |

## 7 CONCLUSION

In this work, we propose MOAI, an efficient and secure transformer inference framework on encrypted data. MOAI leverages consistent matrix packing strategies, rotation-free Softmax and LayerNorm evaluations, and interleaved batching, removing format conversion, minimizing HE rotations, achieving substantial speedups and outperforming the state-of-the-art HE-based transformer inference work THOR Moon et al. (2024)(CCS'25). The proposed techniques are applicable to other revised transformer-based architectures, like Powerformer Park et al. (2024b) (ACL'25), to speedup the secure inference time.

## 8 REPRODUCIBILITY STATEMENT

Our implementation is publicly available via the anonymous GitHub repository linked in the submission. All datasets used in our experiments are public. Parameter settings, libraries, models, datasets, and hardware are detailed in Section 6.

The execution time of THOR and Powerformer in Table 4 and Table 5 are taken from their paper, which are tested in the same hardware environment as ours (a single NVIDIA A100 GPU). More details can be found in Section 6.2. The execution time of NEXUS in Table 8 are obtained by invoking its Softmax and LayerNorm in its open-source implementation. The GPU execution time of THOR in Table 8 are obtained from their paper, which is tested in the same hardware environment as ours (a single NVIDIA A100 GPU). THOR does not provide an open-source CPU implementation nor report CPU results, so we omit the CPU time for THOR. More details can be found in Appendix H.1.

With these resources, all reported results in this work are reproducible.

### ACKNOWLEDGMENTS

This work is a result of the joint lab collaboration between Ant International and Nanyang Technological University, established as part of a five-year partnership to advance digital trust and privacy-enhancing technologies (PETs), with funding provided by Ant International. This research is supported by the National Research Foundation, Singapore and Infocomm Media Development Authority under its Trust Tech Funding Initiative. Any opinions, findings and conclusions or recommendations expressed in this material are those of the author(s) and do not reflect the views of National Research Foundation, Singapore and Infocomm Media Development Authority.

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

## APPENDIX

## A  USAGE OF LARGE LANGUAGE MODELS

ChatGPT is used only for improving grammar and wording in draft preparation. Authors reviewed and revised all content and take full responsibility for the intellectual content of the submission.

## B  EXTENDED PRELIMINARIES: THE BERT MODEL

We define the mean and variance of a vector as functions from $\mathbb{R}^n \to \mathbb{R}$. We denote the softmax function and layer normalization as functions from $\mathbb{R}^n \to \{0, 1\}^n$, specifically:

$$\text{softmax}(\mathbf{x}) = \left[ \frac{\exp(x_i)}{\sum_{r \in [n]} \exp(x_r)} \right]_{i \in [n]} \qquad \text{LayerNorm}(\mathbf{x}) := \left[ \gamma \cdot \frac{x_i - \text{mean}(\mathbf{x})}{\sqrt{\text{var}(\mathbf{x})}} + \beta \right].$$

where $\gamma, \beta$ are model parameters. The functions $\text{softmax}$ and $\text{LayerNorm}$ can be directly extended to matrices, by applying the functions to each *row* of the input matrix.

**Embedding.** Let the input has $m$ tokens. First it is embedded to $m$ vectorized token embeddings, each of which has dimension $d$. We denote the it as matrix $X_e \in \mathbb{R}^{m \times d}$. $d$ is also defined as the

*hidden size* of the model. Let $P \in \mathbb{R}^{m \times d}$ be the positional embeddings for the input tokens. Then, the embedding input is obtained by computing $X := X_e + P$.

**Multi-Head Attention.** Let $X \in \mathbb{R}^{m \times d}$ be the input, $W_Q, W_K, W_V \in \mathbb{R}^{d \times d'}$ be weights matrices, and $\mathbf{b}_Q, \mathbf{b}_K, \mathbf{b}_V$ be bias vectors. An attention block in head $h \in [H]$ is defined as $\text{Attention}(Q, K, V) = \text{softmax}\left(\frac{QK^\intercal}{\sqrt{d'}}\right) V \in \mathbb{R}^{d \times d'}$, where: $Q := XW_Q + \mathbf{1}^\intercal \mathbf{b}_Q$, $K := XW_K + \mathbf{1}^\intercal \mathbf{b}_K$, $V := XW_V + \mathbf{1}^\intercal \mathbf{b}_V$. The multi-head attention is the execution of $H$ parallel attention blocks and we set the $h$-th result to be $X^{(h)}$ where $d' := d/H$. Then the $H$ matrices are concatenated $X^{\text{Concat}} := (X^{(0)} \parallel X^{(1)} \parallel ... \parallel X^{(H-1)}) \in \mathbb{R}^{m \times d}$. The attention block ends with a layer normalization procedure with a residual connection: $X := \text{LayerNorm}(X + X^{\text{Concat}})$.

**Feed Forward Network.** The feed forward network contains four components. A fully connected layer followed by the Gaussian Error Linear Units (GELU) Hendrycks & Gimpel (2016) activation function, another fully connected layer and LayerNorm. The GELU function is defined as $\text{GELU}(x) := \frac{x}{2}\left(1 + \frac{2}{\sqrt{\pi}}\int_0^x e^{-t^2} dt\right)$. Since there is no closed-form expression for GELU, Hendrycks & Gimpel (2016) suggested an approximation by tanh:

$$\text{GELU}(x) \approx 0.5x\left(1 + \tanh\left[\sqrt{2/\pi}\left(x + 0.044715x^3\right)\right]\right). \tag{2}$$

Let $W_{\text{fc1}}, W_{\text{fc2}}$ be weights matrices, and $\mathbf{b}_{\text{fc1}}, \mathbf{b}_{\text{fc2}}$ be bias vectors. The feed forward network can be described as follows: $X := \text{LayerNorm}(X + \text{GELU}(XW_{\text{fc1}} + \mathbf{1}^\intercal \mathbf{b}_{\text{fc1}})W_{\text{fc2}} + \mathbf{1}^\intercal \mathbf{b}_{\text{fc1}}) \in \mathbb{R}^{m \times d}$. The output of feed forward block is set to be the input of the attention block in the next transformer layer. The BERT-base-uncased model Devlin et al. (2019) is using $H = 12$ and it has 12 transformer layers, that is, it contains 12 attention blocks and 12 feed forward blocks.

After 12 transformer layers, the model applies a linear layer with bias and $\tanh(\cdot)$ to the first token ([CLS]) as the aggregated sequence representation for classification tasks. Let this vector be $\mathbf{x}_{[\text{CLS}]} \in \mathbb{R}^d$. The classification layer first computes $\tanh(\mathbf{x}_{[\text{CLS}]}W + \mathbf{1}^\intercal \mathbf{b}) \in \mathbb{R}^d$, denoted as $\mathbf{x}$. Finally let $c$ is the number of classes, the output is $\mathbf{x}W_{\text{cls}} + \mathbf{1}^\intercal \mathbf{b}_{\text{cls}} \in \mathbb{R}^c$.

## C  DETAILS OF MATRIX MULTIPLICATIONS

### C.1  THE BABY-STEP-GIANT-STEP METHOD

In Halevi & Shoup (2021) they optimized the algorithm by baby-step-giant-step method, and the number of HE rotations is reduced to $O(\sqrt{m})$. The idea is to write $m = g \times b$. First for $i \in [b]$, we compute $\text{Rot}_i(\mathbf{v})$ for $b$ times. To compute rotation $\text{Rot}_k(\mathbf{v})$ for $k >= b$, we can write $k = \alpha b + r$ with $r < b$, and

$$\text{rot}_k(\mathbf{v}) := \text{rot}_{\alpha b}\big(\text{rot}_r(\mathbf{v})\big).$$

Combining the above, and with the diagonals of $C$, we will have

$$C\mathbf{v}^\intercal := \sum_{\alpha \in [g]} \text{rot}_{\alpha b}(\sum_{r \in [b]} \text{rot}_{m-\alpha b}\big(\text{Diag}_{\alpha b + r}(C)\big) \otimes \text{rot}_r(\mathbf{v})). \tag{3}$$

### C.2  PSEUDO CODES IN SECTION 3

Algorithm 2 describes the detailed implementation of CPMM when $X$ is encrypted in column packing.

Algorithm 3 describes the detailed implementation for encrypted inputs. For the sake of simplicity, we assume $m = N/2$ in Algorithm 3 and 4 now. Then the rotation of a vector with dimension $m$ can be directly evaluated using one homomorphic rotation. When $m < N/2$ and $m|(N/2)$, Halevi & Shoup (2014) uses sparse packing, i.e., using $m$ slots with index $0, N/(2m), 2 \cdot N/(2m), ..., (m - 1) \cdot N/(2m)$. We will elaborate the our method of full packing in Section 3.2.

### C.3  FURTHER OPTIMIZATION BY HOISTING

Hoisting is a technique to reduce the number of decompositions in HE rotations where the same ciphertext will be rotated multiple times. Halevi & Shoup (2018). More concretely, suppose a

---

**Algorithm 2** Ciphertext-plaintext matrix multiplication $\text{Mult}_{ct,pt}(\text{Enc}_{col}(X), W, B)$.

---

**Input:** $\text{Enc}_{col}(X)$: Column packing of encrypted $X \in \mathbb{R}^{m \times d}$; $W$: Weight matrix with size $d \times d'$.
  $P$: Bias matrix with size $m \times d'$.
**Output:** $\text{Enc}_{col}(XW + B)$: Column packing of encrypted $XW + B \in \mathbb{R}^{m \times d'}$.
 1: Let $Ans := \emptyset$.
 2: Parse $\text{Enc}_{col}(X)$ as $\left\{ \text{Enc}(\mathbf{x}_0^\intercal), \text{Enc}(\mathbf{x}_1^\intercal), ..., \text{Enc}(\mathbf{x}_{d-1}^\intercal) \right\}$.
 3: **for** $j \in [d']$ **do**
 4:    Let $ct_j := (0, 0) \in R_Q^2$ be a naive ciphertext.
 5:    **for** $i \in [d]$ **do**
 6:       $ct_j := \text{Add}\left(ct_j, \text{sMult}(w_{ij}, \text{Enc}(\mathbf{x}_i^\intercal))\right)$.
 7:    **end for**
 8:    Let $\mathbf{p}_j^\intercal$ be the $j$-th column of $P$.
 9:    $Ans := Ans \cup \text{Add}(ct_j, \mathbf{p}_j^\intercal)$.
10: **end for**
11: Return $Ans$.

---

**Algorithm 3** Ciphertext-ciphertext matrix multiplication: $\text{Mult}_{col,col}(\text{Enc}_{col}(Q), \text{Enc}_{col}(K))$.

---

**Input:** $\text{Enc}_{col}(Q), \text{Enc}_{col}(K)$: Column packing of encrypted matrices $Q, K \in \mathbb{R}^{m \times d'}$.
**Output:** $\text{Enc}_{diag}(QK^\intercal)$: Diagonal packing of encrypted matrix $QK^\intercal \in \mathbb{R}^{m \times m}$.
 1: Let $Ans := \emptyset$.
 2: Parse $\text{Enc}_{col}(Q)$ as $\left\{ \text{Enc}(\mathbf{q}_0^\intercal), \text{Enc}(\mathbf{q}_1^\intercal), ..., \text{Enc}(\mathbf{q}_{d'-1}^\intercal) \right\}$.
 3: Parse $\text{Enc}_{col}(K)$ as $\left\{ \text{Enc}(\mathbf{k}_0^\intercal), \text{Enc}(\mathbf{k}_1^\intercal), ..., \text{Enc}(\mathbf{k}_{d'-1}^\intercal) \right\}$.
 4: **for** $j \in [m]$ **do**
 5:    Let $ct_j := (0, 0) \in R_Q^2$ be a naive ciphertext.
 6:    **for** $i \in [d']$ **do**
 7:       **if** $j > 0$ **then**
 8:          $temp := \text{Rot}_j^{\text{HE}}(\text{Enc}(\mathbf{k}_i^\intercal))$.
 9:       **else**
10:          $temp := \text{Enc}(\mathbf{k}_i^\intercal)$
11:       **end if**
12:       $ct_j := \text{Add}\left(ct_j, \text{Mult}(\text{Enc}(\mathbf{q}_i^\intercal), temp)\right)$.
13:    **end for**
14:    $Ans := Ans \cup ct_j$.
15: **end for**
16: Return $Ans$.

---

---

**Algorithm 4** Ciphertext-ciphertext matrix multiplication: $\text{Mult}_{diag,col}(\text{Enc}_{diag}(C), \text{Enc}_{col}(V))$.

---

**Input:** $\text{Enc}_{diag}(C)$: Diagonal packing of encrypted matrices $C \in \mathbb{R}^{m \times m}$. $\text{Enc}_{col}(V)$: Column packing of encrypted matrix $V \in \mathbb{R}^{m \times d'}$.

**Output:** $\text{Enc}_{col}(CV)$: Column packing of encrypted matrix $CV \in \mathbb{R}^{m \times d'}$.

1: Let $Ans := \emptyset$.
2: Parse $\text{Enc}_{diag}(C)$ as $\left\{ \text{Enc}(\text{Diag}_0(C)), \text{Enc}(\text{Diag}_1(C)), ..., \text{Enc}(\text{Diag}_{m-1}(C)) \right\}$. Let $c_i := \text{Enc}(\text{Diag}_i(C))$, for $i \in [m]$.
3: Parse $\text{Enc}_{col}(V)$ as $\left\{ \text{Enc}(\mathbf{v}_0^\intercal), \text{Enc}(\mathbf{v}_1^\intercal), ..., \text{Enc}(\mathbf{v}_{d'-1}^\intercal) \right\}$.
4: Let $b := \lceil \sqrt{m} \rceil$, $g := \lceil m/b \rceil$.
5: **for** $j \in [d']$ **do**
6:    **Baby step:** For $i \in [b]$, let $\beta_i := \text{Rot}_i^{\text{HE}}(\text{Enc}(\mathbf{v}_j^\intercal))$.
7:    **Giant step:** Compute the ciphertext of $C\mathbf{v}_j^\intercal$:

$$ct_j := \sum_{\alpha \in [g]} \text{Rot}_{\alpha b}^{\text{HE}} \left( \sum_{r \in [b]} \text{Mult}(\text{Rot}_{m-\alpha b}^{\text{HE}}(c_{\alpha b + r}), \beta_r) \right).$$

8:    $Ans := Ans \cup ct_j$.
9: **end for**
10: Return $Ans$.

---

ciphertext is rotated $N$ times and let $m$ denote the length of the gadget vector. Without hoisting, performing $N$ HE rotations requires $N$ automorphism, $N$ decomposition and $N$ key switching operations. With hoisting, the decomposition is performed only once and reused for all rotations, replacing the remaining $(N-1)$ decompositions with $(Nm-N)$ additional automorphisms. Because decomposition is more expensive than automorphism, hoisting yields a efficiency improvement.

Now we report the efficiency improvements that can be obtained through hoisting. In summary, hoisting can accelerate the amortized time about at most $2.29$ seconds out of the total amortized time cost of $141.3$ seconds, i.e., an improvement of $1.62\%$. In MOAI, hoisting can accelerate rotations in ciphertext-ciphertext matrix multiplications. We tested the time cost of each operation in CKKS scheme using SEAL library SEAL. The ratio of time for *automorphism:decomposition:key switching* is roughly equal to $1 : 40 : 80$. Therefore the portion of the improvement from hoisting is $\frac{(N+40N+80N)-(mN+40+80N)}{N+40N+80N} = \frac{41-m}{121} - \frac{40}{121N} < \frac{41-m}{121}$. In SEAL library, $m$ is usually set in the range of $10-20$ in secure inference of transformers, which indicates a $16\% - 26\%$ improvement when the same ciphertext is rotated multiple times.

Further, for fair comparison with existing works Moon et al. (2024); Park et al. (2024b); Zhang et al. (2024a), which do not indicate using the hoisting technique, all our comparisons (e.g., Table 1, Table 2 and Table 7) and experimental results (e.g., Table 3, Table 4, Table 5, Table 9 and Table 11) do not use hoisting.

## D    DETAILS OF INTERLEAVED BATCHING

### D.1    NAIVE BATCHING AND INTERLEAVED BATCHING

Suppose we are given $N/(2m)$ vectors $\{\mathbf{x}^{(r)} \in \mathbb{R}^m\}_{r \in [N/(2m)]}$:

$$\mathbf{x}^{(0)} = [x_0^{(0)}, x_1^{(0)}, ..., x_{m-1}^{(0)}], ..., \mathbf{x}^{(N/(2m)-1)} = [x_0^{(N/(2m)-1)}, x_1^{(N/(2m)-1)}, ..., x_{m-1}^{(N/(2m)-1)}],$$

the naive idea of batching is to arrange them one by one (e.g., in Zhang et al. (2024a)):

$$\bar{\mathbf{x}} := [\mathbf{x}^{(0)}, \mathbf{x}^{(1)}, ..., \mathbf{x}^{(N/(2m)-1)}] = [x_0^{(0)}, ..., x_0^{(1)}, ..., x_0^{(N/(2m)-1)}, ..., x_{m-1}^{(N/(2m)-1)}] \in \mathbb{R}^{N/2}.$$

We call $\bar{\mathbf{x}}$ the naive batching of $N/(2m)$ vectors $\{\mathbf{x}^{(r)} \in \mathbb{R}^m\}_{r \in [N/(2m)]}$. For example, consider batching $[x_0, x_1, x_2]$ and $[y_0, y_1, y_2]$. The naive batching generates $[x_0, x_1, x_2, y_0, y_1, y_2]$, while the interleaved batching generates $[x_0, y_0, x_1, y_1, x_2, y_2]$. This naive batching method is compatible with the ciphertext-plaintext matrix multiplication (Algorithm 2), but it is not compatible with the

ciphertext-ciphertext matrix multiplications (Algorithm 3 and Algorithm 4). More concretely, this naive batching method is not compatible with rotation. When "rotate" the batched vector (e.g., Line 8 of Algorithm 3), the desired output should be a ciphertext encrypting the following "internal rotations":

$$[\text{Rot}_j(\mathbf{k}^{(0)}), \text{Rot}_j(\mathbf{k}^{(1)}), ..., \text{Rot}_j(\mathbf{k}^{(N/(2m)-1)})] .$$

In other words, it should be rotating each $\mathbf{k}^{(i)}$ "internally", which is different from directly rotating the batched vector by $\text{Rot}_j(\bar{\mathbf{k}})$. Moon et al. (2024) proposes a method to obtain internal rotation, using 2 normal CKKS rotations and 2 plaintext-ciphertext multiplications. As a toy example, consider batching $[x_0, x_1, x_2]$ and $[y_0, y_1, y_2]$. To "internally" left rotate by 1, we should use the naive batching $\bar{\mathbf{x}} := [x_0, x_1, x_2, y_0, y_1, y_2]$ to generate $[x_1, x_2, x_0, y_1, y_2, y_0]$. This can be done by $\text{Rot}_1(\bar{\mathbf{x}}) \otimes [1, 1, 0, 1, 1, 0] = [x_1, x_2, 0, y_1, y_2, 0]$. Then $\text{Rot}_{-2}(\bar{\mathbf{x}}) \otimes [0, 0, 1, 0, 0, 1] = [0, 0, x_0, 0, 0, y_0]$. By summing them together we obtain $[x_1, x_2, x_0, y_1, y_2, y_0]$.

### D.2 PROOF OF LEMMA 3.2

*Proof.* On the one hand, by Eq. (1), we have

$$\text{Rot}_{jN/(2m)}(\tilde{\mathbf{x}}) := [x_j^{(0)}, x_j^{(1)}, x_j^{(2)}, ..., x_j^{(N/(2m)-1)},$$
$$x_{j+1}^{(0)}, x_{j+1}^{(1)}, x_{j+1}^{(2)}, ..., x_{j+1}^{(N/(2m)-1)},$$
$$...$$
$$x_{j-1}^{(0)}, x_{j-1}^{(1)}, x_{j-1}^{(2)}, ..., x_{j-1}^{(N/(2m)-1)}] .$$

On the other hand, $\forall r \in [N/(2m)]$, we replace $\mathbf{x}^{(r)}$ in Eq. (1) by $\text{Rot}_j(\mathbf{x}^{(r)}) = [x_j^{(r)}, x_{j+1}^{(r)}, ..., x_{j-1}^{(r)}]$. Then we also obtain

$$\text{Rot}_{jN/(2m)}(\tilde{\mathbf{x}}) := [x_j^{(0)}, x_j^{(1)}, x_j^{(2)}, ..., x_j^{(N/(2m)-1)},$$
$$x_{j+1}^{(0)}, x_{j+1}^{(1)}, x_{j+1}^{(2)}, ..., x_{j+1}^{(N/(2m)-1)},$$
$$...$$
$$x_{j-1}^{(0)}, x_{j-1}^{(1)}, x_{j-1}^{(2)}, ..., x_{j-1}^{(N/(2m)-1)}] .$$

This finishes the proof. $\square$

### D.3 INTERLEAVED BATCHING FOR MATRICES

Similarly, suppose we are given $N/(2m)$ matrices $\{X^{(r)} \in \mathbb{R}^{m \times d}\}_{r \in [N/(2m)]}$. Let $\tilde{X} \in \mathbb{R}^{\frac{N}{2} \times d}$ be the *interleaved batching* matrix of $\{X^{(r)} \in \mathbb{R}^{m \times d}\}_{r \in [N/(2m)]}$. The $i$-th column of $\tilde{X}$ is $\tilde{\mathbf{x}}_i^\intercal \in \mathbb{R}^{N/2}$, which is the interleaved batching vector of **all** the $i$-th column of each $X^{(r)}$, for $r \in [N/(2m)]$, according to Eq. (1) [7]:

$$\tilde{\mathbf{x}}_i^\intercal := [x_{i,0}^{(0)T}, x_{i,0}^{(1)T}, x_{i,0}^{(2)T}, ..., x_{i,0}^{(N/(2m)-1)T},$$
$$x_{i,1}^{(0)T}, x_{i,1}^{(1)T}, x_{i,1}^{(2)T}, ..., x_{i,1}^{(N/(2m)-1)T}, \tag{4}$$
$$...$$
$$x_{i,m-1}^{(0)T}, x_{i,m-1}^{(1)T}, x_{i,m-1}^{(2)T}, ..., x_{i,m-1}^{(N/(2m)-1)T}] .$$

Now we are can formally define the column packing and the diagonal packing for the interleaved batching.

**Definition D.1** (Column Packing for Interleaved Batching). *Suppose we are given $N/(2m)$ matrices $\{X^{(r)} \in \mathbb{R}^{m \times d}\}_{r \in [N/(2m)]}$, the interleaved batching matrix of them is $\tilde{X} \in \mathbb{R}^{\frac{N}{2} \times d}$, whose $i$-th column is given by Eq. (4), for $i \in [d]$. Then the column packing for $\tilde{X}$ is a set of $d$ CKKS ciphertexts, denoted as:*

$$\text{Enc}_{col}(\tilde{X}) := \{ \text{Enc}(\tilde{\mathbf{x}}_0^\intercal), \text{Enc}(\tilde{\mathbf{x}}_1^\intercal), ..., \text{Enc}(\tilde{\mathbf{x}}_{d-1}^\intercal) \} ,$$

*where $\tilde{\mathbf{x}}_i^\intercal \in \mathbb{R}^{N/2}$ is the $i$-th column of $\tilde{X}$. We say that $\text{Enc}_{col}(\tilde{X})$ is the **interleaved column packing** of encrypted $\{X^{(r)} \in \mathbb{R}^{m \times d}\}_{r \in [N/(2m)]}$.*

---

[7] Here, we use $x_{i,j}^{(r)T}$ to represent the $j$-th entry of vector $\mathbf{x}_i^{(r)T}$.

**Definition D.2** (Diagonal Packing for Interleaved Batching). *Suppose we are given $N/(2m)$ square matrices $\{X^{(r)} \in \mathbb{R}^{m \times m}\}_{r \in [N/(2m)]}$. Define:*

$$\widetilde{diag}(X) := \{\widetilde{diag}(X)_i \in \mathbb{R}^{N/2}\}_{i \in [m]} \, ,$$

*where $\widetilde{diag}(X)_i$ is the interleaved batching vector given by Eq. (1) using all the $i$-th (upper) diagonals of $X^{(r)}$, for each $r \in [N/(2m)]$. The (upper) diagonal packing of is a set of $m$ CKKS ciphertexts, denoted as:*

$$\mathrm{Enc}_{diag}(\widetilde{diag}(X)) := \left\{ \mathrm{Enc}(\widetilde{diag}(X)_0), \mathrm{Enc}(\widetilde{diag}(X)_1), ..., \mathrm{Enc}(\widetilde{diag}(X)_{m-1}) \right\} \, .$$

*We use $\widetilde{diag}(X)_i$ as a **row** vector. We say that $\mathrm{Enc}_{diag}(\widetilde{diag}(X))$ is the **interleaved diagonal packing** of encrypted $\{X^{(r)} \in \mathbb{R}^{m \times m}\}_{r \in [N/(2m)]}$.*

We conclude that the column/diagonal packing for interleaved batching is compatible with all the three homomorphic matrix multiplication algorithms (Algorithm 2, 3 and 4) using Lemma 3.2. Note that all the input and output ciphertexts of these algorithms will be encrypting the vectors in the form of the interleaved batching. Algorithm 5, 6 and 7 are the modified algorithms for homomorphic matrix multiplications using the interleaved batching.

For the HE evaluations of softmax (Algorithm 1), layer normalization ( 8) and GELU, we assert that the algorithm with interleaved batching input remains the same. This is because in these algorithms, the rows of the input matrix are independently evaluated. In the HE evaluations of softmax and layer normalization, if we add a permutation of rows of the input matrix, there will be the same permutation of rows of the output matrix. Note that interleaved batching is effectively a permutation of rows of the matrix. Therefore, Algorithm 1 and 8 remains the same when the input is in interleaved batching.

---

**Algorithm 5** $\mathrm{Mult}_{ct,pt}(\mathrm{Enc}_{col}(\tilde{X}), W, P)$ using interleaved batching.

---

**Input:** $\mathrm{Enc}_{col}(\tilde{X})$: Interleaved column packing of encrypted $\{X^{(r)} \in \mathbb{R}^{m \times d}\}_{r \in [N/(2m)]}$; $W$: Weight matrix with size $d \times d'$. $P$: Bias matrix with size $m \times d'$.
**Output:** $\mathrm{Enc}_{col}(\tilde{X}W + \tilde{P})$: Interleaved column packing of encrypted $\{X^{(r)}W + P \in \mathbb{R}^{m \times d'}\}_{r \in [N/(2m)]}$.
 1: Let $Ans := \emptyset$.
 2: Parse $\mathrm{Enc}_{col}(\tilde{X})$ as $\left\{ \mathrm{Enc}(\tilde{\mathbf{x}}_0^{\mathsf{T}}), \mathrm{Enc}(\tilde{\mathbf{x}}_1^{\mathsf{T}}), ..., \mathrm{Enc}(\tilde{\mathbf{x}}_{d-1}^{\mathsf{T}}) \right\}$.
 3: Let $\tilde{P}$ be the interleaved batching matrix of $N/(2m)$ repeated matrices $P$.
 4: **for** $j \in [d']$ **do**
 5:     Let $ct_j := (0,0) \in R_Q^2$ be a naive ciphertext.
 6:     **for** $i \in [d]$ **do**
 7:         $ct_j := \mathrm{Add}\left(ct_j, \mathrm{sMult}(w_{ij}, \mathrm{Enc}(\tilde{\mathbf{x}}_i^{\mathsf{T}}))\right)$.
 8:     **end for**
 9:     Let $\tilde{\mathbf{p}}_j^{\mathsf{T}} \in \mathbb{R}^{N/2}$ be $j$-th column of $\tilde{P}$, i.e., the interleaved batching vector of $N/(2m)$ repeated $j$-th column of $P$.
10:     $Ans := Ans \cup \mathrm{Add}(ct_j, \tilde{\mathbf{p}}_j^{\mathsf{T}})$.
11: **end for**
12: Return $Ans$.

---

## E    COMPLEXITY COMPARISON

We provide a detailed comparison of the complexity of our matrix multiplications with prior FHE-based secure transformer inference works: NEXUS Zhang et al. (2024a)(NDSS'25), THOR Moon et al. (2024)(CCS'25), and Powerformer Park et al. (2024b)(ACL'25).

We focus on the number of key switches (KS), which refers the total count of relinearizations in ciphertext-ciphertext multiplication, rotations, conjugations and any other operations requiring a KS. The number of KS is counted in one transformer layer in the BERT-base model.

In Table 7, we provide a detailed comparison of the number of key switches required in the BERT-base model inference. For NEXUS, results are taken from Zhang et al. (2024a) and Table 4 in Moon

---

**Algorithm 6** $\text{Mult}_{col,col}(\text{Enc}_{col}(\tilde{Q}), \text{Enc}_{col}(\tilde{K}))$ using interleaved batching.

---

**Input:** $\text{Enc}_{col}(\tilde{Q}), \text{Enc}_{col}(\tilde{K})$: Interleaved column packing of encrypted $\{Q^{(r)} \in \mathbb{R}^{m \times d'}\}_{r \in [N/(2m)]}$ and encrypted $\{K^{(r)} \in \mathbb{R}^{m \times d'}\}_{r \in [N/(2m)]}$, respectively.

**Output:** $\text{Enc}_{diag}(\widetilde{diag}(QK^{\intercal}))$: Interleaved diagonal packing of encrypted $\{Q^{(r)}(K^{(r)})^{\intercal} \in \mathbb{R}^{m \times m}\}_{r \in [N/(2m)]}$.

1: Let $Ans := \emptyset$.
2: Parse $\text{Enc}_{col}(\tilde{Q})$ as $\left\{ \text{Enc}(\tilde{\mathbf{q}}_0^{\intercal}), \text{Enc}(\tilde{\mathbf{q}}_1^{\intercal}), ..., \text{Enc}(\tilde{\mathbf{q}}_{d'-1}^{\intercal}) \right\}$.
3: Parse $\text{Enc}_{col}(\tilde{K})$ as $\left\{ \text{Enc}(\tilde{\mathbf{k}}_0^{\intercal}), \text{Enc}(\tilde{\mathbf{k}}_1^{\intercal}), ..., \text{Enc}(\tilde{\mathbf{k}}_{d'-1}^{\intercal}) \right\}$.
4: **for** $j \in [m]$ **do**
5:     Let $ct_j := (0,0) \in R_Q^2$ be a naive ciphertext.
6:     **for** $i \in [d']$ **do**
7:         **if** $j > 0$ **then**
8:             $temp := \text{Rot}_{jN/(2m)}(\text{Enc}(\tilde{\mathbf{k}}_i^{\intercal}))$.
9:         **else**
10:            $temp := \text{Enc}(\tilde{\mathbf{k}}_i^{\intercal})$
11:         **end if**
12:         $ct_j := \text{Add}\left(ct_j, \text{Mult}(\text{Enc}(\tilde{\mathbf{q}}_i^{\intercal}), temp)\right)$.
13:     **end for**
14:     $Ans := Ans \cup ct_j$.
15: **end for**
16: Return $Ans$.

---

**Algorithm 7** $\text{Mult}_{diag,col}(\text{Enc}_{diag}(\widetilde{diag}(C)), \text{Enc}_{col}(\tilde{V}))$ using interleaved batching.

---

**Input:** $\text{Enc}_{diag}(\widetilde{diag}(C))$: Interleaved diagonal packing of encrypted $\{C^{(r)} \in \mathbb{R}^{m \times m}\}_{r \in [N/(2m)]}$.
    $\text{Enc}_{col}(\tilde{V})$: Interleaved column packing of encrypted $\{V^{(r)} \in \mathbb{R}^{m \times d'}\}_{r \in [N/(2m)]}$.

**Output:** $\text{Enc}_{col}(\widetilde{CV})$: Interleaved column packing of encrypted $\{C^{(r)}V^{(r)} \in \mathbb{R}^{m \times d'}\}_{r \in [N/(2m)]}$.

1: Let $Ans := \emptyset$.
2: Parse $\text{Enc}_{diag}(\widetilde{diag}(C))$ as $\left\{ \text{Enc}(\widetilde{diag}(C)_0), \text{Enc}(\widetilde{diag}(C)_1), ..., \text{Enc}(\widetilde{diag}(C)_{m-1}) \right\}$. Let $c_i := \text{Enc}(\widetilde{diag}(C)_i)$, for $i \in [m]$.
3: Parse $\text{Enc}_{col}(\tilde{V})$ as $\left\{ \text{Enc}(\tilde{\mathbf{v}}_0^{\intercal}), \text{Enc}(\tilde{\mathbf{v}}_1^{\intercal}), ..., \text{Enc}(\tilde{\mathbf{v}}_{d'-1}^{\intercal}) \right\}$.
4: Let $b := \lceil \sqrt{m} \rceil$, $g := \lceil m/b \rceil$.
5: **for** $j \in [d']$ **do**
6:     **Baby step:** For $i \in [b]$, let $\beta_i := \text{Rot}_{iN/(2m)}(\text{Enc}(\tilde{\mathbf{v}}_j^{\intercal}))$.
7:     **Giant step:** Compute the encrypted interleaved batching vector of $\{C^r(\mathbf{v}_j^{(r)})^{\intercal}\}_{r \in [N/(2m)]}$:

$$ct_j := \sum_{\alpha \in [g]} \text{Rot}_{\alpha b N/(2m)}\left( \sum_{r \in [b]} \text{Mult}(\text{Rot}_{(m-\alpha b)N/(2m)}(c_{\alpha b + r}), \beta_r) \right).$$

8:     $Ans := Ans \cup ct_j$.
9: **end for**
10: Return $Ans$.

et al. (2024), and normalized by batch size. [8] The results of THOR are from Table 4 in Moon et al. (2024). For Powerformer, equations are derived from Section C.2 and Algorithm 14, with numbers from Tables 11 and 12 in Park et al. (2024b). According to Algorithm 14 in Park et al. (2024b), Powerformer requires several BlockTrans, BlockSigma and BlockTau operations before each ciphertext–ciphertext matrix multiplication BlockMult'. All the 3 operations involve $\Theta(\sqrt{k})$ HE rotations (i.e., key switches in this case). The execution time breakdown in Table 5 further validates the results in Table 7. To evaluate all matrix multiplications in a single layer of transformer model, MOAI achieves the least number of key switches.

Table 7: Comparison of HE matrix multiplications in transformer architecture. The concrete parameters are based on BERT-base-uncased model: $m = 128$, $d = 768$, $d' = 64$, $t = 3072$, $H = 12$. $N$ is the CKKS ring dimension. $c = 16$ is from THOR. $k = 64$ is from Powerformer.

| Function | Scheme | Key switch Equation | Number |
|---|---|---|---|
| $\{XW_*^{(h)}\}_{h\in[H]}$ $X \in \mathbb{R}^{m\times d}, W_* \in \mathbb{R}^{d\times d'}$ $* = Q, K, V$ | NEXUS | $2md/(N/(2m))$ | 768 |
| | THOR | $\frac{m}{2} + 3(H-2)\frac{d'}{c} + \frac{3d'}{c}$ | 192 |
| | Powerformer | $\approx \sqrt{\frac{5}{9}} \cdot 2\sqrt{\frac{md(3d)}{N/2}}$ | 122 |
| | **MOAI (ours)** | 0 | **0** |
| $\{Q^{(h)}(K^{(h)})^\intercal\}_{h\in[H]}$ $Q^{(h)} \in \mathbb{R}^{m\times d'}$ $K^{(h)} \in \mathbb{R}^{m\times d'}$ | NEXUS | $m(d'+1)H/(N/(4m))$ | 780 |
| | THOR | $\frac{m}{2c}(\frac{m}{2}+7) + \frac{m}{8}(\log c + 3) + \frac{m}{2}$ | 460 |
| | Powerformer | $\approx (17 + 14\sqrt{2})\sqrt{k} + 3k$ | 486 |
| | **MOAI (ours)** | $((md'+m)/(N/(2m))) \cdot H$ | **390** |
| $\{\text{softmax}(Q^{(h)}(K^{(h)})^\intercal)V^{(h)}\}_{h\in[H]}$ $\text{softmax}(Q^{(h)}(K^{(h)})^\intercal) \in \mathbb{R}^{m\times m}$ $V^{(h)} \in \mathbb{R}^{m\times d'}$ | NEXUS | N/A | N/A |
| | THOR | $m(\log c + 3)/4 + m\frac{m+6}{4c}$ | 492 |
| | Powerformer | $\approx (19 + 10\sqrt{2})\sqrt{k} + 3k$ | 457 |
| | **MOAI (ours)** | $d'(\sqrt{m} + m + \sqrt{d'})/(N/(2m)) \cdot H$ | **444** |
| $(X^{(0)} \| ... \| X^{(H-1)})W_{fc}$ $X^{(h)} \in \mathbb{R}^{m\times d'}$ $W_{fc} \in \mathbb{R}^{d\times d}$ | NEXUS | $2d^2/((N/(4m)))$ | 9216 |
| | THOR | $\frac{d'}{2} + 2(\frac{H}{2} - 1)\frac{m}{c}$ | 122 |
| | Powerformer | $\approx \sqrt{\frac{2}{3}} \cdot 2\sqrt{\frac{md^2}{N/2}}$ | 75 |
| | **MOAI (ours)** | 0 | **0** |
| $XW_{fc1}$ $X \in \mathbb{R}^{m\times d}, W_{fc1} \in \mathbb{R}^{d\times t}$ | NEXUS | $(2md)/(N/(4m))$ | 1536 |
| | THOR | $\frac{m}{2} + \frac{2t}{c}$ | 448 |
| | Powerformer | $\approx \sqrt{\frac{1}{2}} \cdot 2\sqrt{\frac{mdt}{N/2}}$ | 135 |
| | **MOAI (ours)** | 0 | **0** |
| $XW_{fc2}$ $X \in \mathbb{R}^{m\times t}, W_{fc2} \in \mathbb{R}^{t\times d}$ | NEXUS | $(2mt)/(N/(4m))$ | 6144 |
| | THOR | $\frac{m}{2} + \frac{2d}{c}$ | 160 |
| | Powerformer | $\approx \sqrt{\frac{1}{2}} \cdot 2\sqrt{\frac{mtd}{N/2}}$ | 132 |
| | **MOAI (ours)** | 0 | **0** |
| Total (one transformer layer) | NEXUS | - | > 18444 |
| | THOR | - | 1874 |
| | Powerformer | - | 1407 |
| | **MOAI (ours)** | - | **834** |

# F DETAILS OF NON-LINEAR EVALUATIONS

## F.1 LAYERNORM DETAILS

In this section we give the detailed implementation of $\text{LayerNorm}(X)$. The overview of the flow is shown in Figure 4. Recall that for matrix input, LayerNorm is a "row-to-row"" function applied to each row of the input. Given one row $\mathbf{x} = [x_i]_{i \in [d]}$ from $X$, the $i$-th term of $\text{LayerNorm}(\mathbf{x})$ is $\gamma\frac{x_i - \text{mean}(\mathbf{x})}{\sqrt{\text{var}(\mathbf{x})}} + \beta$. After computing $\text{mean}()$ and $\text{var}()$ we can obtain two different forms of layerNorm

---

[8]NEXUS does not specify how it evaluates $\text{softmax}(Q^{(h)}(K^{(h)})^\intercal)V^{(h)}$, so we exclude this result for NEXUS.

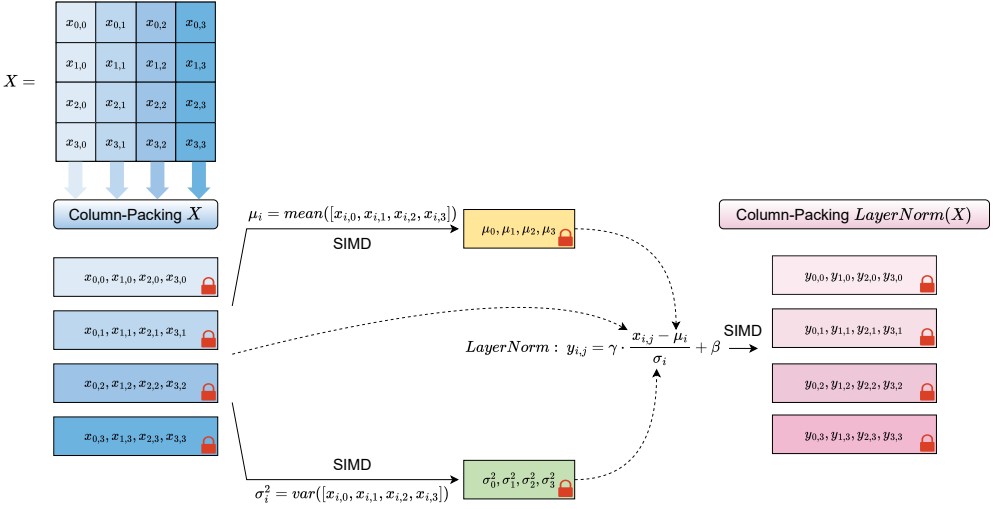

Figure 4: The flow of HE evaluation of LayerNorm.

where $S := \sum_i x_i$:

$$\frac{\gamma}{\sqrt{d}} \frac{dx_i - S}{\sqrt{\sum_i (dx_i - S)^2/d^2}} + \beta \ . \tag{5}$$

$$\frac{\gamma}{d} \frac{dx_i - S}{\sqrt{\sum_i (dx_i - S)^2/d^3}} + \beta \ . \tag{6}$$

We give the detailed implementation in Algorithm 8. The parameter $t \in \{1, 2\}$ means to choose Equation 5 or Equation 6. It is for the higher precision of the inverse square root $1/\sqrt{x}$. After attention block, we choose $t = 1$ and compute LayerNorm according to Equation 5. After feed forward block, we choose $t = 2$ and compute LayerNorm according to Equation 6.

**Remark.** We point out the problem of the HE LayerNorm implementation of NEXUS Zhang et al. (2024a). Comparing with our Equation 5 and 6, they propose:

$$\sqrt{d}\gamma \cdot \frac{dx_i - S}{\sqrt{\sum_i (dx_i - S)^2}} + \beta \ .$$

Note that the input to the inverse square root evaluation is $\sum_i (dx_i - S)^2$, which is about $\Theta(d^3)$. In the BERT-base-uncased model $d = 768$, which makes $\sum_i (dx_i - S)^2$ extremely large and is out of the supported data range. Therefore their algorithm is unable to compute LayerNorm in BERT-base-uncased model.

## F.2 GELU APPROXIMATION

We approximate $\tanh(\cdot)$ in Equation 2 using a degree-23 polynomial $P_{23}(x)$ on interval $[-20, 10]$ which is enough in our experiment. The approximation method is the famous Remez algorithm Remez (1934). It is noteworthy to point out that all polynomial-approximation method only support a specific interval. As suggested by Cheon et al. (2022), a degree-$\Omega(R)$ polynomial is required when using Remez or other minimax approximation with fixed maximum error on the domain interval $[-R, R]$. If the model needs larger input interval, we apply the *domain extension polynomial* $B_r(x) := \frac{x}{L^r} - \frac{4}{27B^2L^{3r}}x^3$ from Cheon et al. (2022). In their algorithm, let $P(x)$ be the approximation polynomial of $\tanh(x)$ on $[-B, B]$. Then $P \circ B_0(x)$ can take input from a larger interval $[-LB, LB]$ for proper $L$. (Theorem 4 in Cheon et al. (2022)). This can be further extended to input interval $[-L^s B, L^s B]$, by using $P \circ B_{s-1} \circ ... \circ B_0(x)$.

---

**Algorithm 8** HE evaluation of LayerNorm.

---

**Input:** $\text{Enc}_{col}(X)$: Column packing of encrypted $X \in \mathbb{R}^{m \times d}$; model parameter $\gamma, \beta \in \mathbb{R}$; algorithm type $t \in \{1, 2\}$.

**Output:** $\text{Enc}_{col}(\text{LayerNorm}(X))$: Column packing of encrypted $\text{LayerNorm}(X) \in \mathbb{R}^{m \times d}$.

1: Let $Ans := \emptyset$.
2: Parse $\text{Enc}_{col}(X)$ as $\big\{ \text{Enc}(\mathbf{x}_0^\intercal), \text{Enc}(\mathbf{x}_1^\intercal), ..., \text{Enc}(\mathbf{x}_{d-1}^\intercal) \big\}$.
3: Compute $\text{Enc}(d \cdot \text{mean}(X))$, where $\text{mean}(X) \in \mathbb{R}^m$ is a vector containing the mean of each row in $X$:
$$\text{Enc}(d \cdot \text{mean}(X)) := \sum_{i \in [d]} \text{Enc}(\mathbf{x}_i^\intercal) ,$$

i.e., the $i$-th element in $\text{mean}(X)$ is the mean of the $i$-th *row* in X.
4: Compute $\text{Enc}(d^3 \cdot \text{var}(X))$, where $\text{var}(X) \in \mathbb{R}^m$ is a vector containing the variance of each row in $X$:
5: **for** $j \in [d]$ **do**
6:     Compute $tmp := \text{sMult}(d, \text{Enc}(\mathbf{x}_i^\intercal)) - \text{Enc}(d \cdot \text{mean}(X))$.
7:     Let $tmp_j := \text{Mult}(tmp, tmp)$.
8: **end for**Let $\text{Enc}(d^3 \cdot \text{var}(X)) = \sum_j tmp_j$.
9: **if** $t = 1$ **then**
10:     Compute $\text{ct} := \text{sMult}(\frac{1}{d^2}, \text{Enc}(d^3 \cdot \text{var}(X)))$.
11:     Use the Goldschmidt division algorithm Goldschmidt (1964); Qu & Xu to obtain the inverse square root ($1/\sqrt{x}$) of ct, denoted as $\text{ct}_{sqroot}$.
12:     Compute $\text{ct}_1 := \text{sMult}(\frac{\gamma}{\sqrt{d}}, \text{ct}_{sqroot})$.
13: **end if**
14: **if** $t = 2$ **then**
15:     Compute $\text{ct} := \text{sMult}(\frac{1}{d^3}, \text{Enc}(d^3 \cdot \text{var}(X)))$.
16:     Use the Goldschmidt division algorithm Goldschmidt (1964); Qu & Xu to obtain the inverse square root ($1/\sqrt{x}$) of ct, denoted as $\text{ct}_{sqroot}$.
17:     Compute $\text{ct}_2 := \text{sMult}(\frac{\gamma}{d}, \text{ct}_{sqroot})$.
18: **end if**
19: **for** $i \in [d]$ **do**
20:     $tmp_i := \text{Mult}\left(\text{ct}_t, (\text{sMult}(d, \text{Enc}(\mathbf{x}_i^\intercal)) - \text{Enc}(d \cdot \text{mean}(X)))\right) + \beta.$
21:     $Ans := Ans \cup tmp_i$.
22: **end for**
23: Return $Ans$.

---

Very recently, Rho et al. (2024) proposed another method to control the data range. They pre-fix the desired data range of the output of each component in the transformer. Then they design punish function accordingly and perform pre-training to the model weights.

We also point out that using the piecewise polynomial in NEXUS Zhang et al. (2024a) to approximate GELU has several problems. NEXUS uses a different HE evaluation of GELU which is based on the line of MPC-based solution Lu et al. (2023); Dong et al. (2023). More concretely, NEXUS calculate GELU ($\mathbb{R} \to \mathbb{R}$) by the following piecewise polynomial:

$$\text{GELU}(x) := \begin{cases} 0 & x \leq -4 \\ P_3(x) & -4 < x \leq -1.95 \\ P_6(x) & -1.95 < x \leq 3 \\ x & x > 3 \end{cases} \tag{7}$$

Here $P_3(x)$ and $P_6(x)$ are polynomials with degree 3 and 6.

First, NEXUS reports the supported input range is $[-8, 8]$. We find that this range is not enough for BERT-base-uncased model, and in our experiment the range is $[-20, 10]$. Next, their piecewise polynomial relies heavily on determining which interval does the encrypted $x$ fall in. This is done by HE evaluation of the an *approximated* sign function. For encrypted number $x$, NEXUS uses 4 encrypt bits $b_1(x), b_2(x), b_3(x), b_4(x)$ so that $b_i(x) = 1$ if and only if $x$ belongs to the $i$-th segment of Equation 7. Ideally, $\text{GELU}(x)$ from Equation 7 can be evaluated by the following for $x \in [-8, 8]$:

$$0 \cdot b_1(x) + P_3(x)b_2(x) + P_6(x)b_3(x) + 3b_4(x) .$$

But in fact, each $b_i$ are approximated bits and it is a small value $\epsilon$ instead of being 0 exactly. This will cause problems. For example, when $x = -8$, in general $P_6(-8)$ is huge, and thus $P_6(-8)b_3(-8) \approx \epsilon P_6(-8)$ is far from 0. What is more, the actually data range we observed is $[-20, 10]$, making $P_6(-20)$ extremely huge. Theoretically one can choose extreme parameters of the CKKS evaluation of sign function so that the error $\epsilon$ is small, but this will increase the time cost dramatically.

### F.3 REMARKS ON OTHER NON-LINEAR FUNCTIONS

As introduced in Sun et al. (2024) and He et al. (2024), we must choose approximations carefully, due to the presence of well-known large activations that are critical for maintaining model performance. Compared with existing works NEXUS Zhang et al. (2024a) and THOR Park (2025), we use a different approximation method based on Cheon et al. (2022) which requires less multiplicative levels. Cheon et al. (2022) allows our solution to be extended to much larger intervals (i.e., wider input range) with very few depth consumption. This also accelerate the non-linear evaluations by starting from a lower ciphertext level, compared with NEXUS and THOR. In addition, our framework is designed in a plug-and-play manner, and it is fully compatible with a substitution of non-linear functions. Any activation function can be supported by our solution by simply adapting the corresponding approximation method.

Another line of works try to replace these non-linear functions in transformer to more FHE-friendly or MPC-friendly ones, such as Park et al. (2024b). A very recent work Jha & Reagen (2025) introduced a layerNorm-free solution utilizing HE and MPC. As we can substitute the non-linear functions, our framework can be easily adopted in Park et al. (2024b) and using ReLU in Jha & Reagen (2025).

## G OPTIMIZED SOFTMAX EVALUATION ALGORITHM

In this section we elaborate the optimized softmax algorithm. Figure 5 shows the error of the Goldschmidt division algorithm. Curves in different colors represent different iterations in the Goldschmidt division algorithm. The y-axis is the absolute error while the x-axis is the real data. The absolute error grows rapidly if the input data is near 0. For stable solution and higher precision, we have to ensure that input data range is not close to 0, and we need at least 10 iterations in the Goldschmidt division algorithm.

Next, at a cost of adding a bootstrapping on only one $\text{ct}_{sum}$ after the third line of Algorithm 1, it can help to reduce the ciphertext modulus in previous matrix multiplications and can significantly reduce the time cost. The detailed optimized HE evaluation of softmax is shown in Algorithm 9.

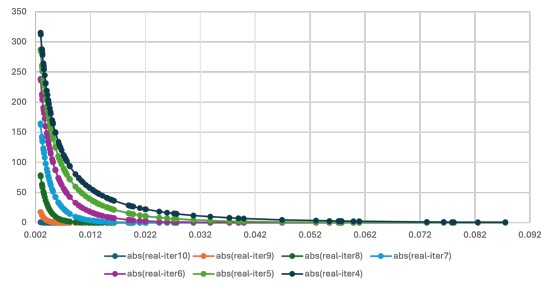 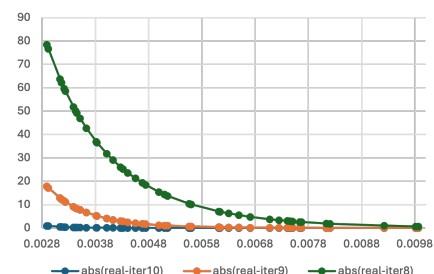

(a) Absolute error of Goldschmidt division algorithm in Softmax

(b) Details of the bottom left corner of the figure

Figure 5: Absolute error of Goldschmidt division algorithm in Softmax

---

**Algorithm 9** HE Softmax evaluation: $\text{Enc}_{diag}(QK^\intercal) \to \text{Enc}_{diag}(\sigma(QK^\intercal))$

**Input:** $\text{Enc}_{diag}(QK^\intercal)$: Diagonal packing of encrypt matrix $QK^\intercal \in \mathbb{R}^{m \times m}$.
**Output:** $\text{Enc}_{diag}(\text{softmax}(QK^\intercal))$: Diagonal packing of encrypt matrix $\sigma(QK^\intercal) \in \mathbb{R}^{m \times m}$.
1: Write $\text{Enc}_{diag}(QK^\intercal)$ as $\{\text{Enc}(\text{Diag}_0(QK^\intercal)), \text{Enc}(\text{Diag}_1(QK^\intercal)), ..., \text{Enc}(\text{Diag}_{m-1}(QK^\intercal))\}$.

2: Evaluate $\text{Enc}(\exp(\text{Diag}_i(QK^\intercal)))$ by SIMD polynomial evaluation $(1 + x/2^r)^{2^r}$.
3: Add ciphertexts to obtain $\text{ct}_{sum} = \text{Enc}(\sum_i \exp(\text{Diag}_i(QK^\intercal)))$.
4: Bootstrap $\text{ct}_{sum}$ to obtain refreshed ciphertext $\text{ct}'_{sum} = \text{Enc}(\sum_i \exp(\text{Diag}_i(QK^\intercal)))$
5: Evaluate HE inverse algorithm on $\text{ct}'_{sum}$ using the Goldschmidt division algorithm Goldschmidt (1964). Let the result be $\text{ct}'_{sum^{-1}} := \text{Enc}(1/\sum_i \exp(\text{Diag}_i(QK^\intercal)))$.
6: For $i \in [m]$, return $\text{Mult}(\text{ct}'_{sum^{-1}}, \text{Enc}(\exp(\text{Diag}_i(QK^\intercal))))$ as the $i$-th component.

---

# H DETAILS OF EXPERIMENTAL RESULTS

## H.1 MICRO-BENCHMARK EVALUATION OF SOFTMAX AND LAYERNORM

We compare our implementation of rotation-free SoftMax and LayerNorm evaluation algorithms with NEXUS Zhang et al. (2024a) (NDSS'25) and THOR Park et al. (2024a) (CCS'25).

All experiments are evaluated with same polynomial ring dimension ($N = 2^{16}$) and slot size ($n = 2^{15}$). All CPU experiments are conducted on an Intel(R) Xeon(R) Platinum 8480+ processor using a single-thread setting, and all GPU experiments are performed on a single NVIDIA A100 GPU. We evaluate NEXUS using its open-source implementation[9], invoking its SoftMax and LayerNorm modules in both CPU and GPU environments. In contrast, THOR does not provide an open-source CPU implementation nor report CPU results; therefore, only GPU timings are included for THOR.

The amortized time taken for both functions are given in Table 8. By eliminating all rotations during evaluation, our SoftMax evaluation algorithm achieves at least $4\times$ speedup over NEXUS and $22\times$ over THOR, while our LayerNorm evaluation algorithm achieves at least $87\times$ and $151\times$ speedup, respectively.

Table 8: Comparison between Softmax and LayerNorm implementations

| Scheme | Softmax | | | LayerNorm | | |
|---|---|---|---|---|---|---|
| | Size | CPU time(s) | GPU time(s) | Size | CPU time(s) | GPU time(s) |
| **MOAI** | $128 \times 128$ | **0.62** | **0.005** | $128 \times 768$ | **0.38** | **0.0039** |
| NEXUS (NDSS'25) | $128 \times 128$ | 2.53 | 0.032 | $128 \times 768$ | 46.7 | 0.34 |
| THOR (CCS'25) | $128 \times 128$ | - | 0.11 | $128 \times 768$ | - | 0.59 |

---

[9] https://github.com/zju-abclab/NEXUS

## H.2 INFERENCE TIME ON LLAMA3-8B MODEL

In this section, we apply our methods in LLaMA-3-8B model. LLaMA-3-8B is a 32-layer decoder-only Transformer with 4096 hidden size, 32 attention heads (grouped-query attention), a feed-forward dimension of 14,336 (SwiGLU), RoPE positional embeddings, RMSNorm normalization, and a vocabulary size of 128k tokens.

To evaluate LLaMA-3-8B model, we implement rotation-free softmax with a causal mask and RMSNorm evaluations by using the same idea of rotation-free SoftMax and LayerNorm evaluations. SiLU evaluation, similar to GELU evaluation, uses a polynomial to approximate it according to Ebel et al. (2025). We incorporate NEXUS's ArgMax evaluation algorithm Zhang et al. (2024a) into MOAI to measure total evaluation time. In the extract-token step, we retain only the last token and repack multiple ciphertexts into one, producing row-packed ciphertexts consistent with NEXUS's ArgMax evaluation. The results in Table 9 aggregate microbenchmarks that cover all components required for LLaMA-3-8B inference. The matrix packing strategies remain consistent, without requiring format conversions or transpositions. We use 8-token input for consistency with PUMA Dong et al. (2023) and NEXUS Zhang et al. (2024a). Note that comparing our results with NEXUS is not appropriate because: (1) NEXUS's inconsistent packing formats require costly conversions, which are not reflected in its reported performance, as observed in Moon et al. (2024); Lim et al. (2025); and (2) NEXUS does not implement all LLaMA-3-8B components, such as RoPE and SwiGLU.

Table 9: Amortized breakdown of MOAI encrypted inference of LLaMA-3-8B with 8 input tokens

| Operation in each layer | Description | Depth | (Data dimensions) × (Number of operations) | GPU time(s) |
|---|---|---|---|---|
| Pt-ct MatrixMul | $Q = XW_Q$ | $17 \to 16$ | $(\mathbb{R}^{8 \times 4096} \times \mathbb{R}^{4096 \times 4096})$ | 20.2 |
| Pt-ct MatrixMul | $K = XW_K, V = XW_V$ | $17 \to 16$ | $(\mathbb{R}^{8 \times 4096} \times \mathbb{R}^{4096 \times 1024}) \times 2$ | 10.1 |
| Pt-ct position-wise Mul | RoPE(Q) | $16 \to 15$ | $\mathbb{R}^{8 \times 4096}$ | 0.11 |
| Pt-ct position-wise Mul | RoPE(K) | $16 \to 15$ | $\mathbb{R}^{8 \times 1024}$ | 0.025 |
| Ct-ct MatrixMul | $QK^\mathsf{T}$ | $15 \to 14$ | $(\mathbb{R}^{8 \times 128} \times \mathbb{R}^{128 \times 8}) \times 32$ | 8.6 |
| Casual masked SoftMax | $\sigma'(QK^\mathsf{T})$ | $14 \to 3$ | $\mathbb{R}^{8 \times 8} \times 32$ | 0.6 |
| Ct-ct MatrixMul | $\sigma'(QK^\mathsf{T})V$ | $3 \to 2$ | $(\mathbb{R}^{8 \times 8} \times \mathbb{R}^{8 \times 128}) \times 32$ | 0.2 |
| Pt-ct MatrixMul | $XW_{\text{selfoutput}}$ | $2 \to 1$ | $(\mathbb{R}^{8 \times 4096} \times \mathbb{R}^{4096 \times 4096})$ | 1.3 |
| Bootstrapping | - | $1 \to 20$ | $\mathbb{R}^{8 \times 4096}$ | 8.8 |
| RMSNorm | Normalization | $20 \to 1$ | $\mathbb{R}^{8 \times 4096}$ | 0.06 |
| Bootstrapping | - | $1 \to 20 \to 14$ | $\mathbb{R}^{8 \times 4096}$ | 8.8 |
| Pt-ct MatrixMul | $\mathbf{u} = XW_u, \mathbf{v} = XW_v$ | $14 \to 13$ | $(\mathbb{R}^{8 \times 4096} \times \mathbb{R}^{4096 \times 14336}) \times 2$ | 97.8 |
| SiLU | $\mathbf{h} = \text{SiLU}(\mathbf{u})$ | $13 \to 3$ | $\mathbb{R}^{8 \times 14336}$ | 6.5 |
| Position-wise Mul | $\mathbf{h} = \mathbf{h} \odot \mathbf{v}$ | $3 \to 2$ | $\mathbb{R}^{8 \times 14336}$ | 0.02 |
| Pt-ct MatrixMul | $\mathbf{y} = \mathbf{h}W_o$ | $2 \to 1$ | $(\mathbb{R}^{8 \times 14336} \times \mathbb{R}^{14336 \times 4096})$ | 48.8 |
| Bootstrapping | - | $1 \to 21$ | $\mathbb{R}^{8 \times 4096}$ | 8.7 |
| RMSNorm | Only in the last layer | $21 \to 2$ | $\mathbb{R}^{8 \times 4096}$ | 0.001 |
| Extract token | Only in the last layer | $2 \to 1$ | $(\mathbb{R}^{8 \times 4096} \to \mathbb{R}^{1 \times 4096})$ | 0.07 |
| Bootstrapping | Only in the last layer | $1 \to 21 \to 2$ | $\mathbb{R}^{1 \times 4096}$ | 0.015 |
| Pt-ct Matrix-vector Mul | Only in the last layer | $2 \to 1$ | $(\mathbb{R}^{1 \times 4096} \times \mathbb{R}^{4096 \times 128256})$ | 0.86 |
| Bootstrapping | Only in the last layer | $1 \to 21 \to 11$ | $\mathbb{R}^{1 \times 128256}$ | 0.47 |
| ArgMax | Only in the last layer | $11 \to 1$ | $\mathbb{R}^{128256}$ | 2.5 |
| Total | | | | 224.53 |

**Accuracy** Table 10 reports GLUE benchmark scores under both plaintext and encrypted inputs. The results indicate that the additional error introduced by FHE has only a negligible effect on inference accuracy in LLaMA-3-8B model.

Table 10: Scores on the GLUE benchmarks

| Model | Dataset | SST-2 | QNLI | RTE |
|---|---|---|---|---|
| LLaMA-3-8B | Unencrypted input | 94.9 | 94.2 | 84.2 |
| | Encrypted input | 94.5 | 93.6 | 82.2 |

**Small input length**  We give a experiment result of the small input length, since the decoding usually comes with length 1. We list the time cost in Table 11, testing on LLaMA-3-8B model. For clarity, we omit operations whose runtime contributes only a small portion of the total computation time (e.g., less than 0.1 second).

Meanwhile, integrating some popular techniques from plaintext transformer inference (e.g., KV-cache, token pruning) with FHE is a useful and interesting direction for future work. A very recent work, Zeng et al. (2025) in NeurIPS'25 has integrated secure multi-party computation (MPC) with KV-cache eviction. Zhang et al. introduced token pruning integrated with MPC, which can further reduce the number of tokens in MPC version inference. While up to our knowledge, no prior work has explored FHE versions of the above techniques.

Table 11: Amortized breakdown of MOAI encrypted inference of LLaMA-3-8B with input token length $= 1$

| Operation in each layer | Description | (Data dimensions) $\times$ (Number of operations) | GPU time(s) |
|---|---|---|---|
| Attention Heads | $X \to \sigma'(QK^\intercal)V$ | $\mathbb{R}^{1\times 4096} \to (\mathbb{R}^{1\times 128} \times 32)$ | 1.57 |
| Pt-ct MatrixMul | $XW_{\text{selfoutput}}$ | $(\mathbb{R}^{1\times 4096} \times \mathbb{R}^{4096\times 4096})$ | 0.18 |
| Bootstrapping | - | $\mathbb{R}^{1\times 4096}$ | 1.18 |
| Bootstrapping after LayerNorm | - | $\mathbb{R}^{1\times 4096}$ | 1.15 |
| Pt-ct MatrixMul | $\mathbf{u} = XW_u, \mathbf{v} = XW_v$ | $(\mathbb{R}^{1\times 4096} \times \mathbb{R}^{4096\times 14336}) \times 2$ | 12.22 |
| SiLU | $\mathbf{h} = \text{SiLU}(\mathbf{u})$ | $\mathbb{R}^{1\times 14336}$ | 0.82 |
| Pt-ct MatrixMul | $\mathbf{y} = \mathbf{h}W_o$ | $(\mathbb{R}^{1\times 14336} \times \mathbb{R}^{14336\times 4096})$ | 5.93 |
| Bootstrapping | - | $\mathbb{R}^{1\times 4096}$ | 1.03 |
| ArgMax | Only in the last layer | $\mathbb{R}^{128256}$ | 2.5 |
| Total | | | 26.94 |

### H.3  ERROR ANALYSIS IN ONE TRANSFORMER LAYER

We divide one transformer layer into 4 parts: (1) Row 1 to Row 4 in Table 3. We call it attention layer; (2) Row 5 to Row 8 in Table 3. We call it self-output layer; (3) Row 9 to Row 10 in Table 3. We call it intermediate layer; (4) Row 11 to Row 14 in Table 3. We call it final output layer.

Figure 6 illustrated the absolute errors of evaluating the first layer with 44 input tokens. It shows that the error grows slowly along the evaluation and the medians of them fall into $[0.01, 0.03]$ in the final output of the layer, which is consistent with our error analysis of evaluating 12 layers.

## I  SET UP OPERATIONS OF THE CKKS SCHEME

Finally, we give more details of the CKKS scheme Cheon et al. (2017; 2018). The set up, encryption and decryption of the CKKS scheme can be summerized as follows.

- $Setup(1^\lambda)$. The input is a security parameter $\lambda$. The output is a parameter set $parms$ of the scheme, including the ciphertext modulus $Q$, and polynomial degree $N$.

- $KeyGen(parms)$. The output includes secret key sk, public key pk and other auxiliary data called evaluation keys.

- $\text{Enc}_{\text{pk}}(\mathbf{v})$. Given the vector $\mathbf{v} \in \mathbb{C}^{N/2}$, the output ciphertext is ct $:= (a, b) \in R_Q^2$. For convenience, we will use $\text{Enc}(\mathbf{v})$ when the public key is clear from context. We will use *naive ciphertext* to refer to $(0, 0) \in R_Q^2$ which is a ciphertext of $\mathbf{0}$.

- $\text{Dec}_{\text{sk}}(\text{ct})$. It outputs the vector $\mathbf{v} \in \mathbb{C}^{N/2}$. sk is omitted when it is clear from context.

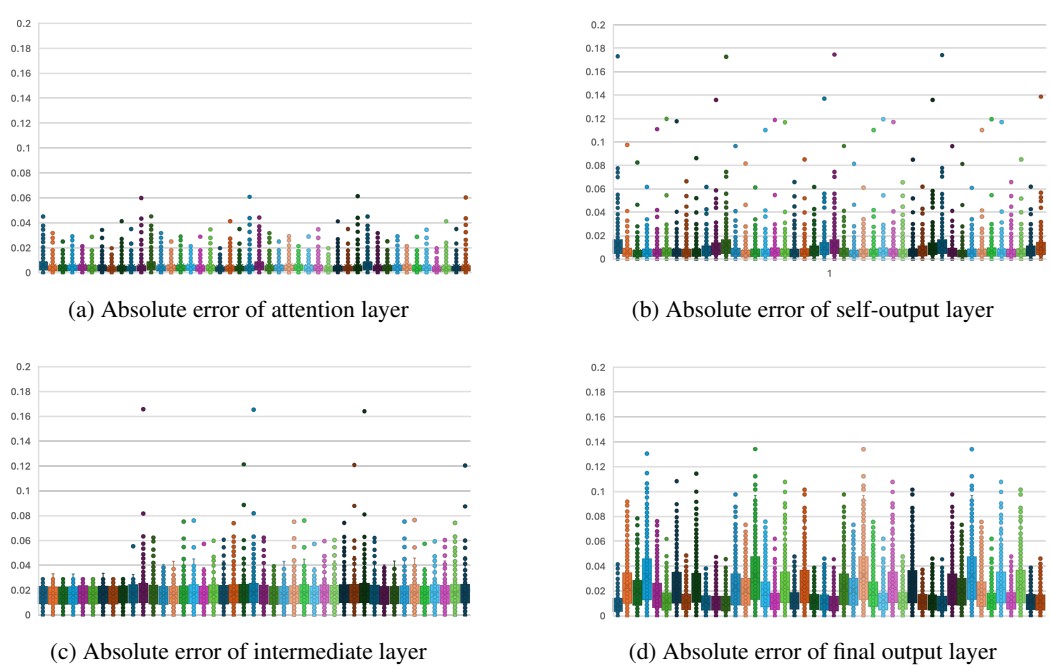

(a) Absolute error of attention layer

(b) Absolute error of self-output layer

(c) Absolute error of intermediate layer

(d) Absolute error of final output layer

Figure 6: Box and whisker figure of absolute error in one layer

