# OpenReview forum: "MOAI: Module-Optimizing Architecture for Non-Interactive Secure Transformer Inference"
_ICLR.cc/2026/Conference — ICLR 2026 Poster_

### Official Review · Reviewer_ycxg · 2025-10-31

**Soundness:** 3
**Presentation:** 3
**Contribution:** 2
**Rating:** 4
**Confidence:** 3

**Summary:**

This paper proposes a new framework named MOAI, designed to address privacy concerns associated with LLM inference when deployed on Cloud Service Providers. The framework is based on Homomorphic Encryption, enabling non-interactive secure inference directly on encrypted data. MOAI introduces three optimizations: an evaluation flow combining column and diagonal packing; "rotation-free" algorithms for the non-linear functions Softmax and LayerNorm; and column packing and interleaved batching techniques. Experimental results claim that MOAI achieves a 52.8% reduction in inference time on the BERT-base model compared to THOR, the current SOTA non-retraining method. Furthermore, its techniques are shown to accelerate the retraining-based SOTA method Powerformer by 55.7%. The paper reports an amortized inference time of 2.36 minutes per input on BERT-base and demonstrates the feasibility of extending the framework to the LLaMA-3-8B model.

**Strengths:**

1.  The paper introduces targeted optimizations for several critical bottlenecks in HE-based Transformer inference, including format conversions, rotations, and matrix multiplications. These optimizations collectively contribute to a significant improvement in the efficiency of secure HE-based inference.
2.  Unlike approaches such as Powerformer, which necessitate model architecture modifications and retraining, MOAI is designed as a "plug-and-play" framework. It does not modify Transformer components, enabling its direct application to existing, pre-trained models and offering superior generality and practical utility.

**Weaknesses:**

Please refer to my questions below.

**Questions:**

While this work presents significant achievements, several key aspects require further clarification from the authors, particularly concerning the method's scalability and its reliability in real-world scenarios.

1.  Scalability Regarding Sequence Length: The paper's introduction suggests application scenarios (e.g., protecting "drafts" or "sensitive records") that may involve long texts. However, all end-to-end experiments in the paper appear to be confined to fixed and relatively short sequence lengths (128 for BERT-base, and only 8 for LLaMA-3-8B).

       Question: The complexity and error propagation of HE operations are typically highly correlated with the sequence length ($m$). Could the authors provide experimental data or analysis illustrating how the MOAI framework's performance (including inference time) and accuracy (cumulative error) would change when processing longer sequences (e.g., $m=1024$ or $m=4096$)? It is currently unclear whether the performance advantages observed on short sequences can be sustained in long-document scenarios.
2.  Accuracy and Task Scope of the LLaMA-3-8B Experiment: Extending the method to LLaMA-3-8B is a significant claim of this paper. However, two major questions surround this experiment:

       Question (Task Scope): The experiment described in Appendix H.2 appears to be a "micro-benchmark" of a single forward pass for an 8-token input, rather than a complete autoregressive generation task. Considering the non-interactive nature of the framework and the 224-second execution time for this single step, the feasibility of this method for practical generative chat applications is questionable.
       Question (Lack of Accuracy Validation): More critically, the paper provides no accuracy evaluation for the LLaMA-3-8B experiment (e.g., Table 9 only reports execution time). All precision and error analyses (e.g., Figure 3 and Table 6) are limited to the 12-layer BERT-base model on classification tasks. Given that LLaMA-3-8B (32 layers) is much deeper than BERT-base (12 layers) and that errors can accumulate layer-by-layer (as shown in Figure 3), how can the authors ensure that the cumulative errors from 32 layers of approximate computations (e.g., for RMSNorm, SiLU, etc.) will not severely degrade the final output quality? We believe the effectiveness of the LLaMA experiment remains ambiguous without this accuracy validation.
3.  Clarity of Application Scenarios (Amortization vs. Latency): The paper's key performance metric (2.36 minutes) is an "amortized time" based on a batch of 256 inputs. This is reasonable for offline analysis tasks.

       Question: Many privacy-sensitive applications (such as secure search) are interactive and more sensitive to "latency" than "throughput." Could the authors provide the latency data for MOAI processing a single input? This would help reviewers more clearly define the practical application scope of the framework.

---

> ### Author Response · Authors · 2025-11-19
> **Response to Reviewer ycxg**
>
> We thank the reviewer for the detailed feedback and thoughtful comments. Below, we respond to the questions and major concerns.
>
> ### Q1: Scalability Regarding Sequence Length
> Our method can naturally support larger token lengths. In fact, the maximum number of tokens per input is determined by the transformer model, rather than by our FHE solution. Based on the FHE parameters we used, each ciphertext can pack up to $2^{15}$ values. Therefore our method can support a token length up to $2^{15}$. For token lengths exceeding this bound, we can simply use additional ciphertexts to store the input. Notably, $2^{15}$ already far exceeds the maximum token lengths of models such as BERT-base (512) and LLaMA-3-8B (8192).
>
> We set token length 128 for BERT-base and 8 for LLaMA-3-8B for our experiments to ensure fair and consistent comparison with prior works [1,2,3]. All the three baselines [1,2,3] reported results on BERT-base with 128-token inputs, and [1] reported experiments on LLaMA-3-8B with 8-token inputs.
>
> ### Q2: Accuracy and Task Scope of LLaMA-3-8B
> Our main contribution and research focus lie in demonstrating the feasibility of FHE-based transformer inference and significantly improving its efficiency. Since all prior works [1, 2, 3] report FHE inference results on BERT-base, we provide a more detailed evaluation on BERT-base to ensure a comprehensive and fair comparison with these baselines.
>
> To show our method is extendible to other transformers, Appendix H.2 gives the breakdown of time cost, ensuring a fair comparison with the state-of-the-art FHE LLaMA-3-8B inference (NEXUS[1]). In addition, we further observe that [1] did not implement all LLaMA-3-8B components, such as RoPE and SwiGLU. We implement these and give a complete inference of LLaMA-3-8B.
>
> Regarding to the accuracy experiment, we are testing it and need more time to finish the test due to high computation overhead of FHE evaluations. We will update the result later. We believe our method has a negligible impact on the inference accuracy of the LLaMA-3-8B model. As shown in Section 6.3, the error increases in the first few layers but then stabilizes, indicating it remains well-bounded in the deeper layers of the model.
>
>
> ### Q3: Clarity of Application Scenarios
> Our solution is designed to reduce the amortized processing time for cases that batching is applied. Batching is useful in several scenarios, such as sending hundreds of FHE-encrypted inputs to a remote text classification system or performing semantic search on a remote server. Our experiments demonstrate that batched inputs achieve higher efficiency compared to prior works [2, 3].
>
> We hope to emphasize that the algorithms for FHE inference are totally different between single-input and batch-input cases. Adapting our approach to a single-input scenario would require a re-design. Prior works typically focus on only one setting: NEXUS [1] and our method target batch-input, while [2, 3] focus on single-input. As shown in our experiments, we can achieve over 50% faster performance than all prior works, which shows the effectiveness of our optimizations in data packing and algorithmic flow.
>
> [1] Zhang, J., Yang, X., He, L., Chen, K., Lu, W. J., Wang, Y., ... & Yang, X. (2025, January). Secure Transformer Inference Made Non-interactive. In NDSS 2025.
>
> [2] Moon, Jungho, et al. "THOR: Secure transformer inference with homomorphic encryption." ACM CCS 2025.
>
> [3] Park, Dongjin, Eunsang Lee, and Joon-Woo Lee. "Powerformer: Efficient and High-Accuracy Privacy-Preserving Language Model with Homomorphic Encryption." Proceedings of the 63rd Annual Meeting of the Association for Computational Linguistics (Volume 1: Long Papers). 2025.

---

> > ### Author Response · Authors · 2025-11-26
> > **Response to Reviewer ycxg part 2**
> >
> > ### Accuracy experiment of LLaMA-3-8B
> > We list the accuracy on the GLUE benchmarks in the following table.
> > The results indicate that the additional error introduced by FHE has only a negligible effect on the inference accuracy in LLaMA-3-8B model.
> >
> > | Model    | Dataset | SST-2 | QNLI | RTE |
> > | -------- | ------- | -------- | ------- | ------- |
> > | LLaMA-3-8B  | Unencrypted input | 94.9% | 94.2% | 84.2% |
> > | LLaMA-3-8B   | Encrypted input     | 94.5% | 93.6% | 82.2% |

---

### Official Review · Reviewer_sUSa · 2025-10-31

**Soundness:** 3
**Presentation:** 2
**Contribution:** 2
**Rating:** 4
**Confidence:** 4

**Summary:**

This paper presents MOAI, a non-interactive secure Transformer inference framework based on homomorphic encryption (HE). The authors propose a new packing strategy aimed at reducing the number of rotations and format conversions, thereby improving efficiency. To eliminate the overhead introduced by format conversion across layers, they propose a consistent matrix packing strategy. Furthermore, the paper presents a rotation-free evaluation scheme for SoftMax and LayerNorm, as well as a rotation-reduced approach for matrix multiplication. Experimental results demonstrate that MOAI improves evaluation time by 52.8% without sacrificing accuracy.

**Strengths:**

1. The topic is relevant and timely. Non-interactive secure Transformer inference is important for privacy-preserving applications in communication-constrained scenarios.

2. The paper is well-structured and easy to follow. Prior techniques such as packing strategies and interleaved batching are explained clearly.

3. The evaluation is thorough, including a detailed breakdown of overheads for individual components. The release of the implementation further improves reproducibility and practical impact.

**Weaknesses:**

1. The technical novelty requires further clarification. The primary contribution appears to lie in the composition and refinement of existing packing strategies. However, several techniques adopted in this paper are not entirely new. For example, reducing the number of rotations via diagonal packing is a well-established approach (e.g., Section 4 in [1]). Applying such a strategy to SoftMax appears to be a straightforward extension.

2. Certain methodological details lack clarity. For instance, in Algorithm 1, the authors describe the evaluation of the SoftMax function, but do not specify the degree of the polynomial used (line 301). Additionally, the authors apply the domain extension technique to approximate the GELU function, but it is unclear why this approach is not similarly applied to the sign function, which would validate the approximation at line 1095. The limitations of previous piecewise approximations are not clearly explained.

3. Some notation and formulations require better consistency. For example, the asymptotic complexities of NEXUS and THOR are essentially the same, yet are presented in different forms. This inconsistency may confuse readers.

**Questions:**

1. Are there experimental results to support the authors' claim about GELU evaluation in Appendix F.2? Specifically, is there quantitative evidence showing accuracy degradation due to the piecewise low-degree polynomial approximation?

2. Given that MOAI is completely non-interactive, could functional bootstrapping [2] be leveraged to reduce the overhead of non-linear functions?

3. Since the Phantom library currently does not support CKKS bootstrapping, and MOAI's GPU implementation is based on Phantom, how is bootstrapping performed in practice?

4. While this work focuses on optimizing FHE evaluation protocols, could techniques from the machine learning side be integrated as well? For example, work elimination [1], token pruning [3], and KV-cache eviction [4] have shown effectiveness in secure Transformer inference. A discussion on the potential integration of such techniques could be beneficial.

[1] Pang, Qi, et al. "Bolt: Privacy-preserving, accurate and efficient inference for transformers.", IEEE S&P 2024.

[2] Alexandru, et al. "General functional bootstrapping using CKKS.", Crypto, 2025.

[3] Zhang, Yancheng, et al. "Cipherprune: Efficient and scalable private transformer inference.", ICLR 2025.

[4] Zeng, Wenxuan, et al. "MPCache: MPC-Friendly KV Cache Eviction for Efficient Private Large Language Model Inference.", NeurIPS 2025.

---

> ### Author Response · Authors · 2025-11-19
> **Response to Reviewer sUSa**
>
> We thank the reviewer for the detailed feedback and thoughtful comments. Below, we respond to the questions and major concerns.
>
> ### 1. Technical novelty
> Compared to SOTA works THOR and Powerformer, our work introduces a fundamentally different packing strategy (including both column-packing and diagonal-packing) that significantly reduces expensive HE rotations while maintaining consistency across the entire evaluation pipeline.  We are the first work to propose rotation-free SoftMax and LayerNorm evaluation algorithms. We outperformed THOR and Powerformer, achieving >50% time cost reduction in the same environment.
>
> ### 2. Methodological clarity
> Thanks for pointing out the unclear items.
>
> In SoftMax evaluation, same as NEXUS, we use r=8 in line 2 in Algorithm 1. Regarding to the number of iterations in the Goldschmidt division algorithm (line 4, Algorithm 1), we analyzed the error introduced in Appendix G (Figure 5) and showed that our choice (10 iterations) is the one with the smallest error. We also discussed the details of multiplicative levels in SoftMax function. Please refer the details in “About bootstrapping” paragraph in Section 5.
>
> Regrading to GELU evaluation, as shown in Section F.2, we provided a detailed explanation of why “sign function + piecewise function” evaluation strategy is not suitable. We showed that a huge error will be introduced when the input is out of the small range (e.g., [-8,8] in NEXUS, which is not enough for BERT-base inference).
>
> ### 3. Formulation consistency
> Thank you for the the suggestion. In table 2 Section 4, we aim to indicate the different algorithms in NEXUS and THOR. (In NEXUS, half of slots are reserved for accumulation among slots, while in THOR, the factor “2” comes from their internal rotations.) We will revise our formulations to achieve better consistency in our next version.
>
> ### Q1: GELU evaluation
> We have analyzed the error introduced at each part of the transformer in Section H.3, including the error from our GELU approximation (see Fig. 6(c)). In addition, Section 6.3 presents both the error of each layer and the corresponding GLUE benchmark results under encrypted inference. Our results show that the approximation of GELU does not introduce significant error, nor does it negatively impact model accuracy.
>
> ### Q2: Functional bootstrapping
> Functional bootstrapping is a technique that can compute a function value f(x) when refreshing the ciphertext. However, we do not adopt this technique for non-linear evaluations for the following reasons:
> 1. In current SOTA functional bootstrapping [2], the input of the function f() is limited to be a number in $\mathbb\{R\}$, instead of a real vector. However, the input of SoftMax and LayerNorm is a vector, so we cannot use functional bootstrapping to evaluate them.
>
> 2. GELU, whose input and output are both in $\mathbb\{R\}$, is suitable for functional bootstrapping. However, the model size limits the efficiency of functional bootstrapping. In our current solution, the bootstrapping before GELU is supposed to bootstrap $128\times768$ slots. If we were to use functional bootstrapping to evaluate GELU, it would need to bootstrap all $128\times3072$ slots. This is significantly slower than in our current solution and would degrade efficiency.
>
> ### Q3: Bootstrapping in Phantom library
> We use the same CKKS bootstrapping implementation as in NEXUS (the link is given in Section G.1) in our GPU implementation. The implementation can be found in `/src/include/source/bootstrapping` in our GPU implementation codebase.

---

> > ### Author Response · Authors · 2025-11-19
> > **Response to Reviewer sUSa part 2**
> >
> > ### Q4: Integration of other techniques
> > We thank reviewer sUSa for commenting the potentially useful techniques in recent publications. First, we hope to clarify why we do not adopt the related techniques. Following prior works on FHE transformer inference [5,6], our focus is converting the original model into a FHE-enabled version. Therefore, our research and [5,6] did not modify the original model’s structure or parameters. The main advantages are to avoid any fine-tuning and enabling a fully “plug-and-play” approach for existing open-source transformers. Reviewers ycxg and qFBv also recognized these advantages. To ensure a fair comparison, we strictly follow the same computation flow used in prior FHE-based transformer works [5, 6, 7].
> >
> > [1] is an MPC-based transformer inference. Prior FHE-based transformer works [6, 7] have already shown superior performance compared to [1] (e.g., Table 4 in [6]). Since our method further outperforms [5,6], it consequently achieves better efficiency than [1] as well. The word-elimination in [1] relies on an "oblivious sorting method" (as noted in [3], [1] becomes computationally expensive due to its reliance on sorting) to remove tokens with lower contributions. However, implementing sorting under FHE is significantly more challenging and costly than in MPC. Efficient FHE-based sorting would indeed make such techniques more practical, but this remains an open challenge.
> >
> > Token pruning in [3] has a similar idea as [1]: reduce the number of tokens. [3] use thresholds to remove "less important" tokens, avoiding sorting all the scores. As [3] is based on ASS, (leveled)HE and OT, it would be an interesting direction to investigate FHE-only token pruning based on thresholds instead of the expensive sorting. [4] is indeed an interesting direction. Since it was published in Neurips’25 which is very close to the ICLR submission deadline, we were unable to incorporate it, but we can explore it further in future work.
> >
> > We once again thank the reviewer for the valuable comments. We hope this addresses the reviewer’s concerns and feel free to raise any additional questions.
> >
> > [1] Pang, Qi, et al. "Bolt: Privacy-preserving, accurate and efficient inference for transformers.", IEEE S&P 2024.
> >
> > [2] Alexandru, et al. "General functional bootstrapping using CKKS.", Crypto, 2025.
> >
> > [3] Zhang, Yancheng, et al. "Cipherprune: Efficient and scalable private transformer inference.", ICLR 2025.
> >
> > [4] Zeng, Wenxuan, et al. "MPCache: MPC-Friendly KV Cache Eviction for Efficient Private Large Language Model Inference.", NeurIPS 2025.
> >
> > [5] Zhang, J., Yang, X., He, L., Chen, K., Lu, W. J., Wang, Y., ... & Yang, X. (2025, January). Secure Transformer Inference Made Non-interactive. In NDSS 2025.
> >
> > [6] Moon, Jungho, et al. "THOR: Secure transformer inference with homomorphic encryption." ACM CCS 2025.
> >
> > [7] Park, Dongjin, Eunsang Lee, and Joon-Woo Lee. "Powerformer: Efficient and High-Accuracy Privacy-Preserving Language Model with Homomorphic Encryption." Proceedings of the 63rd Annual Meeting of the Association for Computational Linguistics (Volume 1: Long Papers). 2025.

---

### Official Review · Reviewer_pF5R · 2025-11-01

**Soundness:** 3
**Presentation:** 3
**Contribution:** 3
**Rating:** 4
**Confidence:** 4

**Summary:**

This work focuses on the secure inference of Transformers when FHE is adopted. The author provides a new flow that includes matrix multiplication and nonlinear functions, mainly to reduce the number of rotations. Results demonstrate the effectiveness compared with THOR and Powerformer.

**Strengths:**

1. The proposed method that allows the output of QKT in diagonal packing is meaningful; this further makes the rotation in softmax removable. Its correctness is true after careful examination and is supposed to guarantee security.

2. The paper has a clear definition and algorithm.

**Weaknesses:**

1. The assumption of batching in Section 3.2 is strange to me. Prior works seldom do this, and I only know the batching occurs during the offline stage. Since batching is not a trivial solution in HE, as one client usually queries only one sentence once and queries from different clients cannot be batched.

2. The proposed method seems to rely on the input with a large sequence length. However, in the widely used GPT, the decoding only uses a sequence length of 1.

3. As reported in prior works (Nexus), the time spent on bootstrapping takes around 2/3 of the total time. But this work does not focus on such a major bottleneck.

**Questions:**

1. The authors use a 23-degree polynomial to approximate the GELU function but do not provide detailed coefficients of the polynomial. The high-degree terms usually have tiny coefficients, such as 10^{-20}. In this case, the fixed-point precision is not enough to express these coefficients and would set them as zero. Given that the high-degree terms are usually extremely large, this will cause a very large error.

2. Could the authors provide the comparison of the #rotation with Nexus when only the QK^T and softmax are included? I believe this can better indicate the optimization of section 3.1. I am also curious about the comparison when batching is not applied.

3. The performance on inputs with small input sequences is more interesting in the GPT model, since the decoding usually comes with an extremely small input length (only 1). Providing a comparison in such cases is important to demonstrate the effectiveness of the proposed method.

---

> ### Author Response · Authors · 2025-11-19
> **Response to Reviewer pF5R**
>
> We thank the reviewer for the constructive feedback and appreciation of our contributions and would like to address the key concerns below:
>
> ### 1. Batching
> Our solution is designed to reduce the amortized processing time for cases where batching is applied. There are some scenarios where batching is common and preferred in encoder-only models like BERT. For example, a sentiment analysis system can batch hundreds of reviews together, and BERT can process them in parallel. Tasks like text classification, semantic search, and document similarity, require a large number of inputs to be grouped and encoded simultaneously to produce a batch of embedding vectors.
>
> Given the substantial computational cost of FHE, prior works [1,2,3] also primarily focus on BERT-base and frequently adopt batching to improve throughput (e.g., Nexus [1] explicitly is evaluated under a batched setting). In these realistic deployment scenarios, reducing the amortized time per input improves batch processing which our framework significantly demonstrates.
>
>
> ### 2. Sequence length
> Our method can naturally support smaller/larger token lengths. Our matrix multiplication algorithms and non-linear function evaluations can be applied in the case where the sequence length is 1. We set token length 128 for BERT-base and 8 for LLaMA-3-8B for our experiments to ensure fair and consistent comparison with prior works [1,2,3]. All the three baselines [1,2,3] reported results on BERT-base with 128-token inputs, and [1] reported experiments on LLaMA-3-8B with 8-token inputs.
>
> However, due to the substantial computational cost of FHE, FHE-based transformer inference is more suitable in offline processing of sensitive inputs instead of online question-answer scenario, where the sequence length is larger and batching several inputs in one processing is preferred. Therefore, we did not consider the case where sequence length is 1 in our work.
>
> If the sequence length is only 1, our solution can batch up to 32768 inputs and evaluating them together to improve throughput, where the amortized time for 32768 inputs is around $0.125\times$ our current result $\approx 28$ seconds in LLaMA-3-8B model. (The minimum valid sequence length in BERT-base is 2 tokens. )
>
> ### 3. Bootstrapping
> Prior works of FHE transformer inference [1,2,3] all reported that bootstrapping is the major bottleneck. However, optimizing the existing CKKS bootstrapping algorithm or designing a novel variant remains highly challenging and is one of the major open problems in the field of cryptography. In fact, [1,2,3] and our manuscript use the same CKKS bootstrapping algorithm.
>
> In our work, we focus on reducing expensive HE operations such as rotations and key switching and designing consistent packing strategies to avoid additional format conversion. These optimizations significantly reduce the runtime outside bootstrapping and thus improve end-to-end performance even under the same bootstrapping scheme.

---

> > ### Author Response · Authors · 2025-11-19
> > **Response to Reviewer pF5R part 2**
> >
> > ### Q1: Error in GELU
> > To reduce the error caused by precision, we follow the method in [4]: first approximate it on a smaller interval using polynomial with smaller degree, and then using domain extension polynomial on larger intervals. This method avoids extremely tiny coefficients.
> >
> > Regarding to error, we discussed the error introduced each part of the transformer in Section H.3 (the error regarding to GELU approximation is shown in Figure 6(c)). We also talked about the error of each layer and the GLUE benchmark scores under encrypted data in Section 6.3. All results shows that our GELU approximation neither introduce large errors nor affect accuracy.
> >
> > ### Q2: Number of rotations in $QK^T$ and SoftMax
> > We listed comparisons of number of rotations which are related to our optimizations in Section 3 in Table 1.
> >
> > To answer Reviewer pF5R's question, we extract the equation for $QK^T$ and SoftMax from Table 1 and Table 2,  and extrapolated with BERT’s parameters (12 layers) into equations. The results are shown in the following table:
> >
> > |          | NEXUS                | NEXUS | MOAI                 |    MOAI    |
> > | ----------- | ----------- | ----------- | ----------- | ----------- |
> > |Operation | Equation (one layer) | Number | Equation (one layer) | Number |
> > |$QK^T$ | $\frac\{md'H\}\{N/4m\}$ | 768 * 12 = 9216 | $\frac\{md'H\}\{N/2m\}$ | 384 * 12 = 4608 |
> > |SoftMax | $\lceil log\_2 m \rceil\frac\{mH\}\{\frac\{N\}\{4\}/2^\{\lceil log\_2 m \rceil\}\}$ | 1008 | 0 | 0 |
> > |Total | - | 10224 | - | 4608 |
> >
> > When only comparing $QK^T$ and SoftMax, our solution saves more than 5000 HE rotations in BERT-base inference. We would like to explain that other than these 2 computations, our solution also reduces the number of rotations in:
> > $XW\_\*, \(\*=Q,K,V\)$, SoftMax$(QK^T)V, XW\_\{fc\_0\}$ , $XW\_\{fc\_1\}$ , $XW\_\{fc\_2\}$ , LayerNorm.
> >
> > These reductions further demonstrate the effectiveness of the optimizations presented in Section 3.1.
> >
> > ### Q3: Effectiveness of solution when input length is 1
> > We thank the reviewer for pointing out the interesting scenario. Our matrix multiplication algorithms and non-linear function evaluations can be applied in the case where the sequence length is 1. In this case, our batching technique supports to batch up to 32768 inputs and evaluate them together to improve throughput. amortized time for 32768 inputs is around $0.125\times$ our current result $\approx 28$ seconds in LLaMA-3-8B model. (The minimum valid sequence length in BERT-base is 2 tokens. ) We are testing it and need more time to finish the test due to high computation overhead of FHE evaluations. We will update the result later.
> >
> > We once again thank the reviewer for the valuable comments. We hope this addresses the reviewer’s concerns and feel free to raise any additional questions.
> >
> > [1] Zhang, J., Yang, X., He, L., Chen, K., Lu, W. J., Wang, Y., ... & Yang, X. (2025, January). Secure Transformer Inference Made Non-interactive. In NDSS.
> >
> > [2] Moon, Jungho, et al. "THOR: Secure transformer inference with homomorphic encryption." ACM CCS 2025.
> >
> > [3] Park, Dongjin, Eunsang Lee, and Joon-Woo Lee. "Powerformer: Efficient and High-Accuracy Privacy-Preserving Language Model with Homomorphic Encryption." Proceedings of the 63rd Annual Meeting of the Association for Computational Linguistics (Volume 1: Long Papers). 2025.
> >
> > [4] Jung Hee Cheon, Wootae Kim, and Jai Hyun Park. Efficient homomorphic evaluation on large intervals. IEEE Transactions on Information Forensics and Security, 17:2553–2568, 2022.

---

> > > ### Author Response · Authors · 2025-11-26
> > > **Response to Reviewer pF5R part 3**
> > >
> > > ### Experimental results for input length 1
> > > The amortized time cost of the case when input length is 1 is about 26.94 seconds.
> > > In summary, we can batch 8 times more data into one ciphertext, resulting in lower amortized time cost.
> > > In addition, when token length is 1, the matrix multiplications like $QK^T$ become inner product, which further simplify the computations.
> > > Please refer to the detailed breakdown time cost in Section H.2 and Table 11 of our revised version.

---

### Official Review · Reviewer_XZwn · 2025-11-01

**Soundness:** 2
**Presentation:** 2
**Contribution:** 1
**Rating:** 2
**Confidence:** 5

**Summary:**

This paper implements BERT inference using a GPU-accelerated implementation of CKKS. The main contributions of this paper are improved analysis of the data packing for both the linear and non-linear operations in encrypted transformer inference. The main benefit of this improved analysis is a reduction in the number of rotations needed for various operations in the transformer inference .

**Strengths:**

I like that they provide their implementation. In general, the experimental results are well-documented.

**Weaknesses:**

The contributions all seem very minor/incremental. There is also no consideration of significant optimizations in the analysis of the rotation counts. These hoisting optimizations, where one ciphertext can be rotated many times at a cost similar to rotating the ciphertext once, can dramatically reduce the runtimes of FHE linear evaluation.

**Questions:**

Not all rotations are created equal. When counting rotations, the number of unique ciphertexts to be rotated often matters much more than the number of output rotations. Did you consider hoisting optimizations when counting rotations?

---

> ### Author Response · Authors · 2025-11-19
> **Response to Reviewer XZwn**
>
> We appreciate the reviewer’s feedback and would like to address the key concerns below:
>
> ### Hoisting in rotations
> We appreciate the reviewer’s suggestion that to consider the hoisting technique in HE rotations where one ciphertext will be rotated many times.
> However, we find that hoisting would not bring significant improvements to our work.
> After implementation and experiments, we find that hoisting can accelerate the amortized time about at most **2.29 seconds** out of the total amortized time cost of 141.3 seconds, i.e., an improvement of at most **1.62%**.
>
> Let us consider rotating a ciphertext N times and let m denote the length of the gadget vector.
>
> |  | Automorphism | Decomposition | Key switching |
> | --- | --- | --- | --- |
> | Without hoisting | N | N | N |
> | With hoisting | Nm | 1 | N |
>
> * Key switching takes up the most time during rotation, as shown in [3,4]. They both included the number of key switching as one effective metric (Table 4 of [3], Table 1 in [4]).
>
> * Hoisting replaces (N-1) decompositions by (Nm-N) automorphisms. In addition, hoisting does not change the number of key switches.
>
> * Therefore, we find that hoisting reduces the time taken for rotations, but to a limited extent.
>
> * Our implementation is based on SEAL (CPU) and PhantomFHE (GPU), as in [2]. SEAL and PhantomFHE did not include hoisting. An author of SEAL also explained that the hoisting technique has limited contribution to the overhead of rotation in SEAL’s implementation in SEAL’s GitHub page.
>
> * Finally, the research works in the line of FHE transformer inference, e.g., [2,3,4], do not use the hoisting technique. For a fair comparison, we do not use hoisting in this manuscript.
>
> We would like to emphasize that our improvements in efficiency is still significant even in the case that hoisting is applied in all existing works.
>
> We tested the time cost of each operation in CKKS scheme using SEAL library. The ratio of time for *automorphism: decomposition : key switching* is roughly equal to 1: 40 : 80. Therefore the portion of the improvement from hoisting is $\frac\{(N+40N+80N)-(mN+40+80N)\}\{N+40N+80N\}=\frac\{41-m\}\{121\}-\frac\{40\}\{121N\}\le \frac\{41-m\}\{121\}$. In SEAL library, m is usually set in the range of 10-20 for our work, THOR, NEXUS and Powerformer, which indicates a 16%-26% improvement in FHE rotations when the same ciphertext is rotated multiple times.
>
> For ciphertext-ciphertext matrix multiplication, both our algorithms and existing works can be accelerated a bit by hoisting. For plaintext-ciphertext matrix multiplication and non-linear functions, where our algorithms are 3-50x faster than THOR[3] and Powerformer[4], our algorithms do not need rotation so hoisting cannot be applied. While existing works could potentially apply hoisting in these components, but the overall improvement achieved through hoisting is very limited as analyzed above. Therefore, our improvement in efficiency of HE-based inference is still significant.
>
> ### Q1: Comparison of rotations
> To achieve a fair comparison with [3,4], we compared both the number of HE rotations and the number of key switching without hoisting. We consider the number of ciphertexts in comparison, and ensure the comparisons are in the same setting. We present in Table 1 Section 3.3, the number of HE rotations. We included in Table 7 Appendix E the comparison between the number of key switches.
>
> We once again thank the reviewer for the valuable comments. We hope this addresses the reviewer’s concerns and feel free to raise any additional questions.
>
> [1] Halevi, Shai, and Victor Shoup. "Faster homomorphic linear transformations in HElib." Annual International Cryptology Conference. Cham: Springer International Publishing, 2018.
>
> [2] Zhang, J., Yang, X., He, L., Chen, K., Lu, W. J., Wang, Y., ... & Yang, X. (2025, January). Secure Transformer Inference Made Non-interactive. In NDSS.
>
> [3] Moon, Jungho, et al. "THOR: Secure transformer inference with homomorphic encryption." ACM CCS 2025.
>
> [4] Park, Dongjin, Eunsang Lee, and Joon-Woo Lee. "Powerformer: Efficient and High-Accuracy Privacy-Preserving Language Model with Homomorphic Encryption." Proceedings of the 63rd Annual Meeting of the Association for Computational Linguistics (Volume 1: Long Papers). 2025.

---

### Official Review · Reviewer_qFBv · 2025-11-08

**Soundness:** 3
**Presentation:** 4
**Contribution:** 3
**Rating:** 6
**Confidence:** 5

**Summary:**

This paper address the latency overhead of  homomorphic encryption (HE) in private LLM inference, in non-interactive setting (HE-only inference, instead of HE+MPC hybrid settings). The Authors have developed MOAI framework to reduce the rotation operation (the most-expensive operation in HE) in key  nonlinearities (e.g., Softmax and LayerNorms) and plaintext-ciphertext matrix multiplication, which reduce the end-to-end private inference latency.

The reduction in number of rotation operations is **substantial**, and also the methodology is very-well explained and it does not required any fine-tuning, and can be adapted to the existing open-sourced LLM.

**Strengths:**

1. Cryptographically secure private LLM inference is an emerging research area but suffers from high latency (in HE-only inference) and communication (in HE+MPC inference) overheads. In HE-only inference latency stems from rotation  and bootstrapping  operations, and the proposed frame work reduces the former to a significant extend.


2. The paper has presented a very-detailed latency breakdown (Table 3) and operation-wise comparison with prior work (Table 1 and Table 2; and Table 4 and Table 5), which is **impressive and very useful** to understand the savings coming from reducing the rotations in linear as well as nonlinear operations). The authors also discuss the pitfalls and other shortcomings in the end-to-end implementation, and the assumptions made, in prior works such as NEXUS and THOR.

3. A detailed discussion and empirical analysis for Softmax approximation is provided in the Appendix G and H.3 (Figure 5 and Figure 6)


4. The approach is simple and relatively straightforward to implement, in a plug-and-play manner. Also, the authors have provided the open-source implementation

**Weaknesses:**

The main limitation of HE-only inference lies in **accurately approximating nonlinear operations**, particularly when compared with hybrid HE+MPC approaches. Polynomial approximations in HE are highly sensitive to the data range over which they are defined. Within a narrow predefined range, the approximation error can remain low, but this restricts the *representational flexibility* of the model. Expanding the range to better capture the activation dynamics requires using *higher-degree polynomials*, which in turn **increases the number of bootstrapping operations** and thus end-to-end latency.

If we want to enforce a tighter data range for constraining activations to the region where the polynomial approximation remains valid, we need to add the regularization terms during training or we need to fine-tuning the model. For example, the polynomial approximation of LayerNorm in HE-only inference [1]


More importantly, this approximating with polynomial approach may not scale well to LLMs, due to the presence of well-known massive activation [2,3] which are critical to maintaining model performance. Consequently, achieving an accurate approximation would require an extremely wide range, which again necessitates a high-degree polynomial---defeating the efficiency goal of HE-only inference.


[1]  Zimerman et al., Converting Transformers to Polynomial Form for Secure Inference Over Homomorphic Encryption, ICML 2024

[2] Sun et al., Massive Activations in Large Language Models, COLM 2024

[3] He at al., Understanding and Minimising Outlier Features in Neural Network Training, NeurIPS 2024

**Questions:**

1. Why GELU latency shown in Table 4 is significantly lower than GELU in THOR, even when a 23-degree polynomial is used for their approximation?

2. If we substitute the GELU with ReLU in FFN, which is shown a promising solution in HE+MPC private inference on LayerNorm-free LLMs [1], then how it would impact the end-to-end performance in HE-only inference ?



[1] Jha et al., AERO: Entropy-Guided Framework for Private LLM Inference

---

> ### Author Response · Authors · 2025-11-19
> **Response to Reviewer qFBv**
>
> We thank the reviewer for the constructive feedback and the appreciation of our contributions and would like to address the key concerns below:
> ### Q1: GELU latency
> Compared to GELU evaluation in THOR, the improvement in our work comes from the following aspects:
>
> 1. Reduced complexity of the GELU approximation. THOR uses a two-step approach to approximate GLU with a composition of two polynomials $f2(f1(x))$, where $f2()$ and $f1()$ are degree-31 and degree-27 respectively. The 2 polynomials should be evaluated sequentially. As discussed in Section 5.2 of THOR, it requires 11 multiplicative levels to evaluate GELU. While our MOAI uses a different approximation method based on [1], which requires less multiplicative levels.  We also designed a parallel polynomial evaluation strategy based on the Paterson–Stockmeyer algorithm, which further accelerates the computation.
> 2. Different starting levels of input ciphertexts. The GELU evaluation in MOAI and THOR starts from different ciphertext levels. The ciphertexts in MOAI starts with 9 multiplicative levels before the GELU evaluation. THOR does not explicitly specify the precise levels for each stage, but since their GELU evaluation consumes 11 levels, the ciphertexts must start with at least 12 levels before the GELU evaluation. The difference in starting level results in the performance gap.
>
> ### Q2: Replace GELU with ReLU
> Thank you for highlighting the promising direction in privacy-preserving inference. In HE-only inference, the difference between using GELU and ReLU is on the complexity of their approximation methods. If a ReLU approximation with lower degree and fewer multiplicative levels is available, it would indeed accelerate the HE-only inference process. Our framework is designed in a plug-and-play manner, and it is fully compatible with such a substitution. Any activation function can be supported by our solution by simply adapting the corresponding approximation method.
>
> We once again thank the reviewer for the valuable comments. We hope this addresses the reviewer’s concerns and please feel free to raise any additional questions.
>
> [1] Jung Hee Cheon, Wootae Kim, and Jai Hyun Park. Efficient homomorphic evaluation on large intervals. IEEE Transactions on Information Forensics and Security, 17:2553–2568, 2022.

---

### Author Response · Authors · 2025-11-26
**General reply to all the reviewers about the revised pdf**

We sincerely thank all the reviewers for their feedback and provide a summary of the primary changes in the revised manuscript. We highlighted our revisions in the updated manuscript.

1.	Further optimization by hoisting.

We thank Reviewer `XZwn` for suggesting considering hoisting technique in HE rotations. After we implement and analyze, we report that in MOAI hoisting can accelerate the amortized time about at most **2.29 seconds** out of the total amortized time cost of 141.3 seconds, i.e., an improvement of **1.62%**. (See Appendix C.3)

2.	Effectiveness of substituting non-linear functions.

We thank Reviewer `qFBv` for highlighting the promising direction in privacy-preserving inference which focuses on replacing non-linear functions in transformer to simpler functions. We analyze that our framework is designed in a plug-and-play manner, and it is fully compatible with a substitution of non-linear (See Appendix F.3)

3.	Accuracy of LLaMA-3-8B.

We thank Reviewer `ycxg` for suggesting adding accuracy experiment in LLaMA-3-8B model. We update GLUE benchmark scores under both plaintext and encrypted inputs in LLaMA-3-8B model. (See Appendix H.2, Table 10)

4.	Effectiveness of solution when input length is 1.

We thank Reviewer `pF5R` for pointing out the scenario. We test our scheme of the small input length on LLaMA-3-8B model and report the time cost. (See Appendix H.2, Table 11)

5.	Other techniques introduced in MPC-based inference.

We thank Reviewer `sUSa` for commenting the potentially useful techniques in recent publications. We add an introduction to the line of works in MPC-based inference. (See Appendix H.2)

---

### Meta-Review · Area_Chair_82VP · 2026-01-06

**Summary:**

Reviewers raised concerns about limited LLaMA-3-8B evaluation, lack of single input latency data, GELU approximation accuracy, batching assumptions, and GPU bootstrapping implementation. One reviewer noted that the proposed method overlooked hoisting but authors showed its runtime impact is minimal. Most concerns are minor clarifications. Given that the proposed method achieves a 50%+ speedup over SOTA and the method remains plug-and-play without requiring retraining, the AC recommends acceptance. However, as the AC is not expert in this area, the decision might not be accurate.

**Reviewer Concerns:**

The rebuttal addressed all major concerns about GELU approximation showing negligible error, batching assumptions and applicability to single input sequences have also been clarified. Hoisting related issue raised by one of the reviewer has been addressed with both analysis and empirical evidence of minimal impact.

**Reviewer Scores:**

I would say almost all reviewers would be positive about this work and probably give 6.

---

### Decision · Program_Chairs · 2026-01-26

Accept (Poster)